# Super-resolution upgrade for deep tissue imaging featuring simple implementation

Patrick Byers[1,2], Thomas Kellerer [1], Miaomiao Li[3,4], Zhifen Chen[3,4], Thomas Huser [2] & Thomas Hellerer[1] ✉

Deep tissue imaging with high contrast close to or even below the optical resolution limit is still challenging due to optical aberrations and scattering introduced by dense biological samples. This results in high complexity and cost of microscopes that can facilitate such challenges. Here, we demonstrate a cost-effective and simple to implement method to turn most two-photon laser-scanning microscopes into a super-resolution microscope for deep tissue imaging. We realize this by adding inexpensive optical devices, namely a cylindrical lens, a field rotator, and a sCMOS camera to these systems. By combining two-photon excitation with patterned line-scanning and subsequent image reconstruction, we achieve imaging of sub-cellular structures in *Pinus radiata*, mouse heart muscle and zebrafish. In addition, the penetration depth of super-resolved imaging in highly scattering tissue is considerably extended by using the camera's lightsheet shutter mode. The flexibility of our method allows the examination of a variety of thick samples with a variety of fluorescent markers and microscope objective lenses. Thus, with a cost-efficient modification of a multi-photon microscope, an up to twofold resolution enhancement is demonstrated down to at least $70\mu m$ deep in tissue.

Super-resolution optical microscopy (SRM) is capable of surpassing the optical diffraction limit and can achieve spatial resolutions well below 200 nm[1]. In the life sciences, this enables the investigation of the finest details of sub-cellular structures (e.g. organelles, viruses, protein complexes, vesicles, etc.)–typically based on fluorescence excitation of specifically labeled target molecules[2–5]. Despite the undeniable advances which were made possible by SRM in cell biology, imaging structures buried deep within biological tissues with super-resolution continues to remain a formidable challenge. SRM techniques based on the localization of single fluorescent molecules can accomplish this, but typically only when combined with some means of optical sectioning such as confocal fluorescence microscopy[6] or oblique plane illumination[7]. In contrast, stimulated emission depletion (STED) microscopy has been demonstrated even for selected in vivo applications[8]. STED often employs high photon dosages to deplete fluorescence from outside the focal spot, which are detrimental to sample viability, or requires special fluorophores in order to work efficiently. Super-resolution structured illumination microscopy (SR-SIM), on the other hand, is a method that provides a twofold resolution improvement, while being gentle enough to allow fast live cell imaging at multiple fluorescent colors simultaneously with conventional fluorophores[9]. We refer to SR-SIM simply as SIM throughout this article for better readability, whereas optical-sectioning SIM is a different method not relevant in this context. In coherent SIM (cSIM) fluorescence in the sample is excited by an interference pattern with a periodicity at or near the diffraction limit[10–12]. By shifting and rotating this pattern across the field of view (FOV), a 2D isotropic increase in resolution across the entire sample can be achieved after computational image reconstruction[13]. The coherent formation of the excitation pattern has a specific

[1]Multiphoton Imaging Lab, Munich University of Applied Sciences, Munich, Germany. [2]Biomolecular Photonics, Department of Physics, Bielefeld University, Bielefeld, Germany. [3]Department of Cardiology, German Heart Center, TUM University Hospital, TUM School of Medicine and Health, Technical University Munich, Munich, Germany. [4]Deutsches Zentrum für Herz- und Kreislaufforschung (DZHK), partner site Munich Heart Alliance (MHA), Munich, Germany. ✉e-mail: hellerer@hm.edu

advantage: interference ensures that the modulation contrast[12] can reach up to 100 percent even at the highest spatial frequency supported by the amplitude transfer function of the objective lens. cSIM is, however, rather limited in its application to deep tissue imaging, due to absorption, phase distortion, and scattering in dense samples which decrease the achievable contrast.

Over the last few years, a number of efforts have been undertaken in order to improve the deep tissue imaging capabilities of SIM. In the majority of cases, however, this involved the construction of complex optical systems that are not easy to implement in most application laboratories. Furthermore, several of these SIM variants that perform better for deep-tissue imaging often do not quite reach the spatial resolution of cSIM. For example, a modest improvement of spatial resolution by a factor of 1.6 at 5 μm imaging depth was reported by an early implementation of sequential line-scanning without compromising excitation modulation contrast[14]. Applying adaptive optics counteracts the deteriorating effects of phase distortion on the excitation modulation contrast which has been shown in[15–19].

SIM was also demonstrated with sequential point-scanning, as it had previously been proposed in image scanning microscopy (ISM)[20], and it was further advanced with all-optical implementations such as OPRA and Re-scan confocal microscopy[21,22]. Here, super-resolution information is extracted from laser-scanning confocal microscopes while maintaining a high signal-to-noise ratio (SNR). ISM in its initial concept demonstrated improvement factors of 1.63 but required long acquisition times[23]. More recently, advanced all-optical implementations significantly increased the temporal resolution to up to 1 Hz but at the expense of resolution improvement factors of only up to 1.44[21,22]. Even faster SIM concepts with resolution improvements close to a factor of two, such as multifocal SIM (MSIM)[24] and instant SIM (iSIM)[25] were then introduced. While imaging 100 μm deep in samples has been demonstated by combining those modalities with two-photon excitation[26], this required a rather complex optical system. Multiview imaging combined with 1D MSIM further suppresses background fluorescence by synchronizing the rolling shutter with the one-photon line-excitation[27]. To truly extend SIM and its variants to even deeper tissue imaging, the power of multi-photon excitation is, however, needed. Recently, living Drosophila melanogaster embryos were imaged by an all-optical approach to ISM with fast-scanning, single-beam two-photon excitation[28]. In an effort to simplify such implementations, striped illumination and camera-based detection in a modified commercial two-photon laser-scanning microscope was demonstrated to achieve super-resolution imaging in plant tissue[29], but also required considerable acquisition times. This short summary shows that there is a need for alternative, easy to implement approaches to super-resolution microscopy based on two-photon excitation to enable deep tissue imaging.

Here, we present a compact and inexpensive optical system, which uses two-photon excited fluorescence, a field rotator, and a sCMOS camera with lightsheet shutter mode (LSS) to allow for image acquisition with enhanced contrast and lateral resolution from deep within fluorescent tissue samples. This system requires very minor modifications of a conventional multi-photon laser-scanning microscope. Further, we provide information about hardware implementation and synchronization, as well as code to control the microscope (Supplementary Section 1). A close to twofold resolution improvement is achieved by striped illumination and field rotation of the excitation and detection paths followed by computational SIM image reconstruction. By using the camera's LSS mode, we find that the detected modulation contrast of the SIM illumination pattern is enhanced even at imaging depths well above 50 μm, which enables deep tissue super-resolution microscopy with ~150 nm spatial resolution. We demonstrate this by imaging structures within *Pinus radiata*, mouse heart muscle, and zebrafish samples.

## Results

### Principles of Lightsheet Line-scanning SIM (LiL-SIM)

A key innovation of LiL-SIM is the exploitation of the camera's LSS mode, which efficiently blocks scattered light as depicted in Fig. 1b. For lightsheet fluorescence microscopy (LSFM) with one-photon excitation, this approach was already published a decade ago giving the name to this special mode of operation[30]. Here, we go a significant step further by applying it to SIM utilizing two-photon excitation, which is not as straightforward as it might seem at first glance. First, the lines of the SIM pattern need to be oriented along three different directions whereas the LSS mode works only in one direction. Second, conventional SIM generates an illumination pattern that is spread over the entire FOV not matching the sequential acquisition required for LSS mode operation and demanding significant laser power for nonlinear excitation. Third, conventional SIM pattern generation via interference is phase-sensitive resulting in a distorted pattern deep inside tissue. On the one hand, applying adaptive optics counteracts the deteriorating effects of phase distortion on the excitation modulation[16–19]. On the other hand, it cannot compensate the negative impact of scattering on the detected modulation which is essential for SIM reconstruction. Our solutions to these obstacles are visualized in Fig. 1a. First, a field rotator placed in the shared excitation and detection beam path allows us to rotate the line focus onto the sample, and will undo this rotation as the signal returns to the camera. Second, we change the conventional excitation pattern based on interference to stepwise scanning of a single line focus. Third, we change the conventional one-photon excitation to two-photon excitation because of its success in deep tissue imaging. We will explain these points in more detail in the following paragraphs numbered (1)–(3).

1) Field rotation is critical for generating illumination patterns under different orientation angles, which then leads to an isotropic resolution enhancement of LiL-SIM. Most commonly, SIM patterns are recorded at three different rotation angles of 0°, 60° and 120°. It is crucial that the different orientations of the illumination pattern and that of the camera's exposure band do not deviate by more than a few milliradians, otherwise the detection efficiency will vary significantly across the FOV due to the poor overlap. This technically demanding task is fulfilled by an optomechanical field rotator - in our case a Dove prism that reverses the rotation of the epi-detected fluorescence signal and adds flexibility by permitting arbitrary rotation angles. In contrast to Abbe-Koenig-prisms, the Dove prism changes the orientation of linear polarization relative to the rotated field[31]. This shortcoming is compensated by mounting a half-wave plate on the rotation stage (see Fig. S7 in the Supplemental Information file for a comparison of the modulation contrast of s- and p-polarized illumination patterns), which has the advantage that all devices of the rotation unit are available off-the-shelf. In this way, the contrast of the illumination pattern is not reduced by depolarization when the light is focused through a high numerical aperture (NA) objective lens. For this purpose, we focus the round laser beam with a cylindrical lens into the back focal plane of the objective and make sure that the linear polarization of the laser is perpendicular to this focal line. By orienting the linear laser polarization perpendicular to the central line-focus at the back aperture we ensure that no depolarization occurs. It should also be noted that mechanical rotation of the Dove prism by an angle $\alpha$ results in an optical field rotation of $2\alpha$. Consequently, if a field rotation of 60° is desired, the Dove prism needs to be rotated by 30°.

2) The final pattern is built up line by line, which has the disadvantage that the excitation modulation decreases with increasing spatial frequency of the pattern. The implications thereof are discussed in detail in section "Discussion". Only this decisive deviation from conventional SIM makes it possible to use the LSS mode of the camera. This improves the detected modulation contrast and thus increases the penetration depth. Furthermore, in comparison to the laser power required for full FOV 2P-illumination, this also reduces the

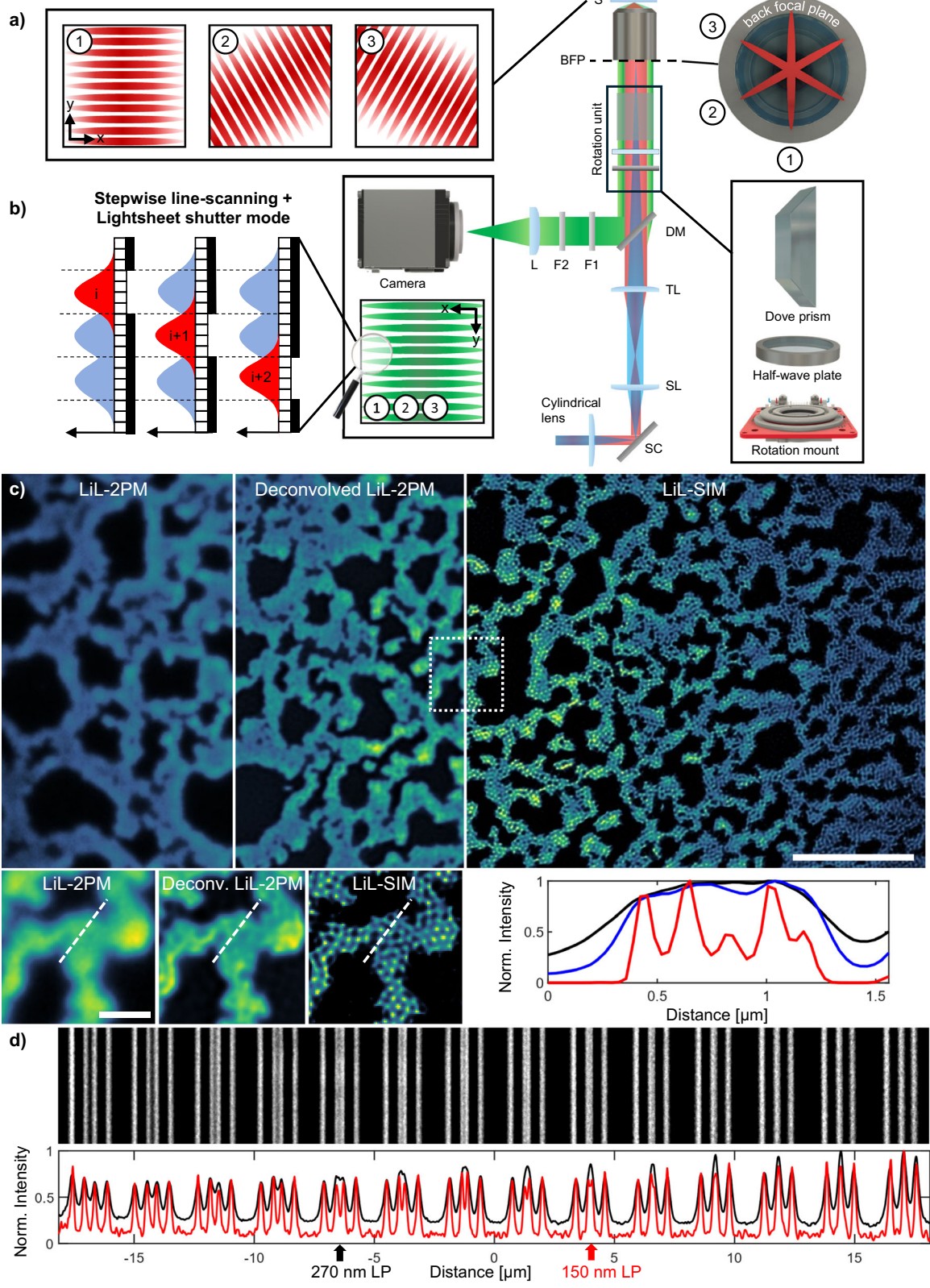

**Fig. 1 | Concept and resolution improvement of lightsheet line-scanning structured illumination microscopy (LiL-SIM). a** Optical schematic of the LiL-SIM setup. The orientation of the excitation line can be set by the rotation unit, which is composed of a piezoelectric rotation stage, a half-wave plate and a Dove prism. SC galvo-scanner, SL scan lens, TL tube lens, DM dichroic mirror, BFP back focal plane, S sample, F1, F2 filters, L lens. **b** The emission of neighboring lines is suppressed with the camera's light sheet shutter mode, leading to less accumulated background compared to rolling shutter mode. **c** Comparison of two-photon excited fluorescence lightsheet line-scanning microscopy (LiL-2PM) (black), deconvolved LiL-2PM reconstruction (blue), and LiL-SIM reconstruction (red) with 190 nm fluorescent beads. **d** Fluorescent line pairs (LPs) can be resolved down to a distance of 270 nm in LiL-2PM (black), while LiL-SIM resolves LPs down to 150 nm (red). The data shown in (**c**) is a representative image out of ($N = 5$) measurements acquired in distinct bead samples. Scale bars: 5 µm, inset 1 µm. The CAD sketch of the rotation mount in this figure is courtesy of Thorlabs, Inc.

required power by a factor given by the number of lines that make up the final pattern (up to 200). Lastly, the pattern spacing can be set by the control voltage of the scanner allowing the greatest possible flexibility in the choice of objective lenses (see Figs. S4 and S5). This desired feature is not common among SIM microscopes that are mostly limited to the use of a single objective lens.

3) The success of two-photon microscopy in deep tissue imaging can be partly explained by the fact that the nonlinear excitation process cuts off the errant waves favoring only the ones that are still in phase leading to a significant amplitude at the focal point. Additionally, the phase distortion and scattering of the fluorescent signal has no detrimental effect in two-photon microscopy as it is integrated over the entire FOV and detected by a point detector rather than a camera. To summarize, in two-photon microscopy, the quality of image formation depends only on the excitation point spread function, exploiting the filtering effect of the nonlinear signal generation process, while only efficient signal collection is important for detection rather than aberration-free imaging. However, integrating the signal is not possible in two-photon SIM because it depends on the fact that the signal is modulated by the sample producing a Moiré pattern which contains the otherwise non-resolvable information. SIM is therefore limited to spatially resolved detection - either simply with a camera or, more advanced, with a single photon counting avalanche photodiode array. We emphasize these points because switching from point detection to camera-based detection in two-photon microscopy degrades image quality, as discussed in subsection "Extending the Penetration Depth with Lightsheet Shutter Mode" and visualized in Fig. 2b.

In conclusion, for generating high modulation contrast at imaging depths exceeding 10 μm, it is of great advantage for two-photon SIM to implement the LSS camera mode together with a Dove prism for field rotation and a half-wave plate for preventing depolarization in high NA objective lenses.

## Achieving super-resolution with SIM

First a general remark: The SIM reconstruction method used in our experiment does not deviate from conventional super-resolution SIM as described in literature[12]. The only exception is the fact that our camera does not have a fixed orientation to the sample but to the illumination pattern because of the implemented field rotation. This requires an additional step called 'digital back rotation' before the standard reconstruction is performed. For all SIM images presented in this paper, five phase shifts were applied to the illumination pattern at each of the three rotation angles (see Fig. 1a). This amounts to a total of 15 images acquired for one reconstructed SIM frame. We have chosen five over three phase shifts because of the resulting higher SNR and reduced artifacts (see Supplementary Fig. S10). For comparison, two other types of images were also acquired and referred to as "LiL-2PM" and "WiL-2PM" throughout this manuscript. LiL-2PM is the abbreviation for lightsheet line-scanning two-photon microscopy and denotes images that are composed of the same 15 raw images as in LiL-SIM (see Fig. 2c). The important difference is that all 15 images are superimposed to eliminate the imprinted pattern and thus making the averaged illumination homogeneous again. This image type is often used as a reference in SIM literature: LiL-SIM and LiL-2PM differ only in resolution, but share the camera's LSS mode. Next, WiL-2PM is the abbreviation for widefield line-scanning two-photon microscopy and denotes images that are recorded with the camera's rolling shutter called here "RS mode" (see Fig. 2b). This also leads to the extinction of the patterned illumination but the cause is different: here, scattered light raises the background thus obscuring the modulation as can be seen by the zebrafish sample in Fig. 2d.

It was not clear from the outset whether the technical precision of the laser scanner would suffice to generate an equidistant illumination pattern with five phase shifts close to the optical resolution limit of the

microscope. Therefore, the galvo-scanner used is equipped with an active feedback loop to outmaneuver the non-linearities inherent in galvo-actuators. Our field test confirms the high phase stability of the illumination patterns over the entire FOV of $67 \times 67 \, \mu m^2$ (see Supplementary Fig. S9). Another important test was performed by imaging a fluorescent slide illuminated with patterns of increasing spatial frequencies (see Supplementary Fig. S12). The test was designed to find the optimum between best possible resolution and sufficient contrast for SIM reconstruction as presented in subsection "Extending the Penetration Depth with Lightsheet Shutter Mode". This crucial test determines the practical resolution limit because for stepwise pattern generation, the modulation contrast depends on the pattern's spatial frequency and the sample's light scattering ability. The theoretical limit and its deviation from conventional SIM will be discussed in section "Discussion". Lastly, we also analyzed if the field rotator introduces aberrations such as field distortion or if it negatively impacts the spatial resolution that can be achieved by this method. As detailed in Supplementary Fig. S8, we found that the field rotator has no negative impact on the imaging performance of the setup except for a slightly decreased signal intensity.

We demonstrated the enhanced spatial resolution by imaging fluorescent beads with a 190 nm diameter illuminated with a 350 nm spaced pattern. As SIM improves the pre-existing resolution of the microscope, it is common to compare the resolution with and without SIM reconstruction - in our case LiL-SIM with LiL-2PM. The LiL-2PM image exhibits a resolution of 275 nm (see Fig. 1c left), both determined by FWHM analysis and decorrelation analysis[32]. The obtained resolution deviates from the theoretical two-photon resolution limit of 212 nm calculated from ref. 33. This comes at no surprise but is typical in two-photon SIM literature (see Supplementary Table T3) and is influenced by optical aberrations and signal-to-background ratio (SBR) among other causes. To quantify the resolution improvement, three post-processing steps were performed by using the open source reconstruction software fairSIM[13]: (1) Richardson-Lucy (RL) deconvolution of the raw images without SIM computation, (2) SIM computation on the raw images without deconvolution, and (3) SIM computation including deconvolution. As mentioned above, the bead FWHM in the averaged LiL-2PM images is 275 nm (see Fig. 1c left). Applying deconvolution without SIM computation results in a bead FWHM of 237 nm (Fig. 1c middle). By SIM computation without deconvolution, the resolution is improved to a bead FWHM of 215 nm (shown in decorrelation curves in Supplementary Fig. S14). Further improvement is achieved by SIM computation including deconvolution, restoring the original bead diameters ranging from 189 to 196 nm (see Fig. 1c right). In the final step, we determined the maximum resolution by measuring fluorescent line pairs (LPs) on a microscope calibration slide. Figure 1d shows a line profile acquired using the 100×/1.49 NA objective lens, where the initial LP spacing is 390 nm, decreasing by 30 nm with each successive pair. This allows for determination of the lateral resolution of the proposed modalities: LiL-2PM resolves LPs down to 270 nm (black), while LiL-SIM achieves an improved resolution of 150 nm (red). Corresponding profiles for the 60×/1.27 NA and 40×/1.15 NA objective lenses are provided in Supplementary Fig. S4.

## Extending the penetration depth with lightsheet shutter mode

Deep tissue imaging benefits from multi-photon excitation and point detection. But SIM requires to replace the point detector by a camera as discussed in subsection "Principles of Lightsheet Line-scanning SIM (LiL-SIM)". Therefore, we performed an experiment to compare the SBR and SNR of point and camera detection which is visualized in Fig. 2a–c. Here, standard point detection is depicted in Fig. 2a and standard camera detection using RS mode in Fig. 2b. The evaluation of SBR and SNR were calculated by dividing the mean of a signal area by the mean of the background, and by the signal standard deviation,

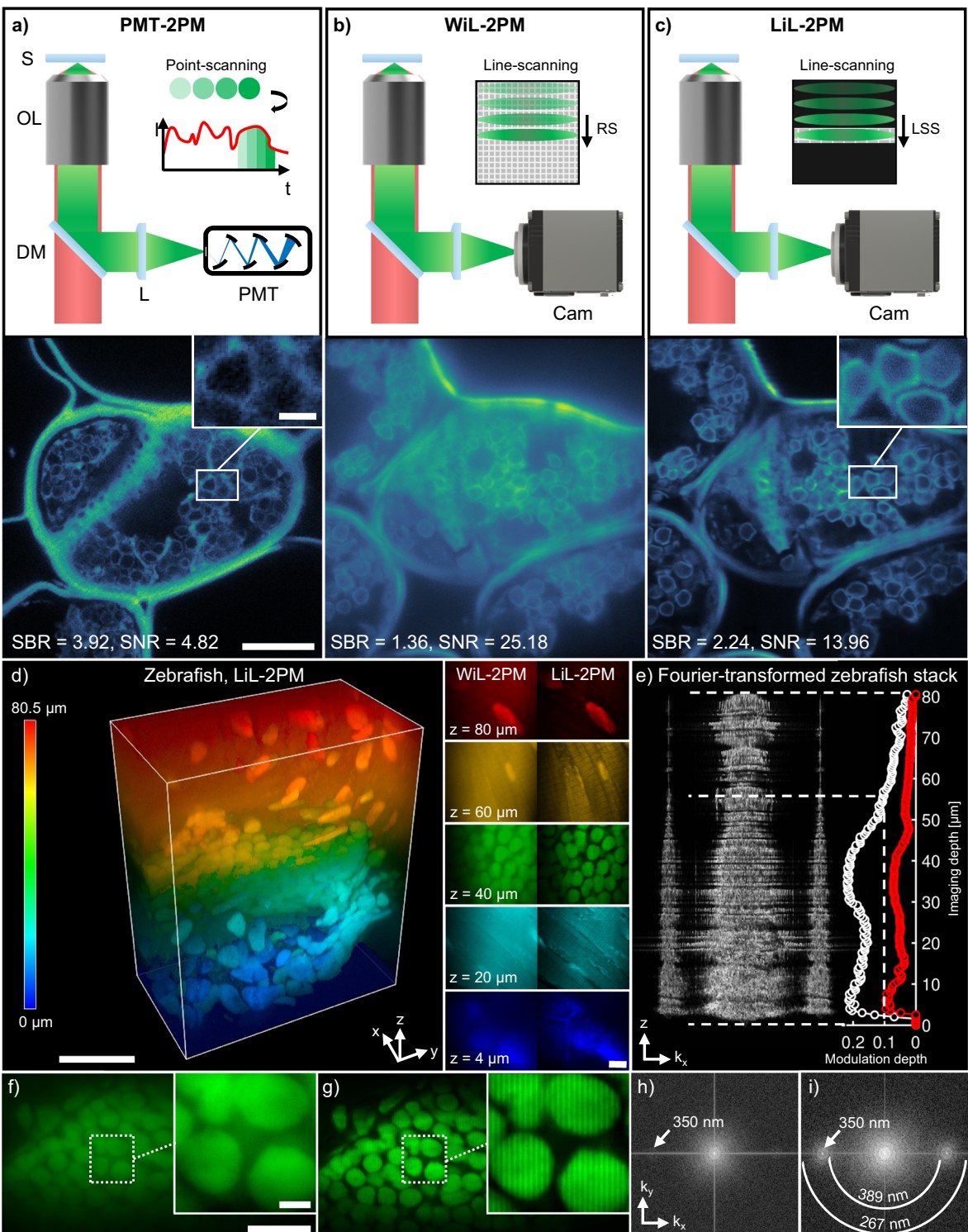

**Fig. 2 | Comparison of two-photon microscopy modalities and evaluation of modulation contrast for LiL-SIM imaging.** Comparison of signal-to-background and signal-to-noise ratio in two-photon fluorescence microscopy images acquired in *Pinus radiata* tissue with **a** photomultiplier tube (PMT), **b** camera with rolling shutter (RS) mode, and **c** camera with lightsheet shutter (LSS) mode. The images presented in (**a**) were acquired with a commercial PMT-based microscope system (PMT-2PM) at the same imaging depth ($z = 10$ μm) but at distinct lateral positions of the specimen. The signal-to-background ratio is increased when using lightsheet line-scanning two-photon microscopy (LiL-2PM) over widefield line-scanning two-photon microscopy (WiL-2PM). **d** Normalized two-photon excited raw LiL-2PM volume of zebrafish with penetration depths ranging from 0 to 80 μm. These raw images were all acquired with the same pattern angle and phase at a pattern spacing of 350 nm. **e** Fourier-transformed planes of the zebrafish stack acquired with LiL-2PM

represent the strength of the modulation contrast of the excitation pattern dependent on the imaging depth. The curve visualizes the strength of the modulation peaks for WiL-2PM (red) and LiL-2PM (white). SIM images can be reconstructed with a modulation contrast higher than 0.1. This corresponds to an imaging depth of 56 μm for super-resolution reconstruction. Extracted planes from 40 μm acquired with **f** WiL-2PM and **g** LiL-2PM demonstrate the improved modulation contrast achieved with LiL-2PM. **h, i** Fourier transforms of the images shown in (**f, g**) visualize the superior modulation contrast achieved with LiL-2PM over WiL-2PM. White rings indicate the spatial frequency in k-space. Data shown in (**a–c**) are representative images taken out of volume stack measurements ($N = 5$ for each modality). The data and the corresponding contrast enhancement shown in (**d–i**) has been verified in ($N = 3$) zebrafish volume stacks taken at distinct locations. Scale bars: **a–c** 10 μm, insets 3 μm. **d** 10 μm, inset 5 μm. **f, g** 10 μm, inset 2 μm.

respectively (further described in Supplementary Fig. S3). While PMT-2PM has the highest SBR, LiL-2PM offers an improvement of almost a factor of 2 compared to WiL-2PM. In terms of SNR, WiL-2PM and LiL-2PM have significantly increased SNR because of longer integration times per pixel (5 ms for WiL- and LiL-2PM vs. 1.16 µs for PMT-2PM). The presented images indicate that only a camera operated in LSS mode (see Fig. 2c) can achieve a comparable image quality to a point detector when imaging is performed in dense specimen layers. Although LSS mode is key for deep tissue imaging, we remind the reader that with SIM the information is buried in the dark areas of the illumination pattern where the signal is raised by the sample's modulation. Standard confocal detection is therefore not a good choice because cutting off the dark areas will result in loss of this information, which then makes super-resolution impossible. This is in contrast to our approach where the exposure band of the camera covers an entire spatial period of the illumination pattern including the dark areas between the lines. Here, we deviate strongly from the above-mentioned line-confocalization in LSFM[30] and also to any other type of confocal detection. Nevertheless, LiL-SIM as presented here is still considered "confocal" because the two-photon excitation limits signal generation to the focal region, which is why multi-photon microscopy is often used for optical sectioning.

What is the advantage of the LSS mode, if not confocalization?— The LSS mode tackles the major issue for SIM that scattering of fluorescence reduces the necessary modulation contrast. All pixels outside of the exposure band are switched off in LSS mode and scattered photons of the neighboring pattern lines are completely suppressed (see Fig. 1b). This is in contrast to earlier approaches to line-scanning SIM, where the interference-modulated line for excitation was scanned across the FOV like a garden rake to generate the pattern[14]. Although this resulted in the excitation modulation being as high as possible, it could not avoid that fluorescence scattering from neighboring pattern lines eradicate the detected modulation contrast. We would like to emphasize this point, because on the one hand the excitation modulation is crucial for resolution improvement, but on the other hand the detected modulation is necessary for SIM computation.

It remains to demonstrate quantitatively how much the modulation contrast is improved by LSS mode compared to RS mode. Therefore, consecutive planes of a zebrafish sample have been recorded with an axial step interval of 0.5 µm (see Fig. 2d). It can be clearly seen in Fig. 2f-g that the modulation contrast at 40 µm depth is still visible with LSS mode (0.2) while it is barely perceptible with RS mode (0.07). The modulation contrast was quantified by Fourier transforming the individual images of the stack followed by the extraction of the maximum peak value from the 2D-power spectrum at the corresponding line pattern frequency (see Fig. 2e). The maximum peak values of all frames are plotted as a function of z to the right of the projected power spectra. An additional dashed line at the value of 0.1 sets a threshold below which artifacts degrade the quality of the reconstructed images. This means that for the zebrafish sample shown in Fig. 2d, the use of RS mode does not even allow the error-free reconstruction of the uppermost image plane at 2 µm depth. Switching to LSS mode, the modulation contrast is increased to an initial value of 0.2, which enables artifact free reconstruction all the way down to an imaging depth at around 56 µm for this particular sample. However, the maximum imaging depth strongly depends on the absorption and scattering properties of the specimen, and can be substantially higher in more homogeneous samples.

## Super-resolved deep tissue imaging with LiL-SIM

In order to demonstrate the power of LiL-SIM, we studied highly scattering tissue samples of *Pinus radiata* by scanning image planes up to a depth of ~30 µm. The integration time per frame was set to 1275 ms, which results in a line time of ~1.25 ms for 1024 lines. The

quality of our results is best confirmed by comparing the imaging modalities WiL-2PM, LiL-2PM and LiL-SIM. Please note that all modalities have the striped two-photon excitation in common. WiL-2PM uses the camera's RS mode, while LiL-2PM uses LSS mode to improve contrast. This can be seen both in 3D renderings (Fig 3a) and in axial center cross-sections (Fig. 3b). Furthermore, the added background rejection and improved contrast are very apparent in the direct side-by-side comparison of WiL-2PM and LiL-2PM with increasing imaging depth provided in Supplementary Movie 1. In Fig. 3c, the super-resolution modality is added: LiL-SIM contains SIM reconstructed images that benefit from the improved contrast, while LiL-2PM consists of superimposed raw images to average out the modulation. On the right are four insets of resin duct cell walls from the cortical region of *Pinus radiata* to illustrate the improved resolution. This is further visualized by two profile plots below that are taken along the dashed lines. Image decorrelation analysis confirms that LiL-SIM almost doubles the lateral resolution, starting at $299 \pm 14$ nm (LiL-2PM) and getting down to $156 \pm 12$ nm (LiL-SIM), determined from ten decorrelation measurements. Fourier-ring correlation (FRC)[34,35], puts the LiL-SIM resolution at $163 \pm 10$ nm (see Fig. 3d, bottom right). Additionally, WiL-2PM is compared to LiL-SIM at different depths to exemplify the overall improvement of our approach (shown in Fig. 3d). It shows the significant difference in quality between conventional camera detection (WiL-2PM) containing substantial blur due to scattered fluorescence and our technique capable of resolving resin canals and cell compartments: LiL-SIM increases not only the resolution beyond the diffraction limit but also obtains better image contrast in deep tissue. The weak honeycomb artifact patterns visible in Fig. 3d ($z = 30$ µm) are likely caused by refractive index mismatches of the immersion medium, coverslip and specimen layers resulting in a decrease of the axial resolution. Please also note that we analyzed the axial resolution of the three different modalities to be $648 \pm 32$ nm for PMT-2PM, $562 \pm 27$ nm for WiL-2PM and $508 \pm 24$ nm for LiL-2PM as well as LiL-SIM, as shown in Supplementary Fig. S6.

An even higher penetration depth is demonstrated by imaging a cross-section of a highly scattering animal tissue sample prepared from mouse heart muscle with actin filaments labeled. While most SIM systems are limited to imaging depths of about 15 µm, we were able to reconstruct image planes up to 70 µm penetration depth without the use of adaptive optics or special camera equipment (shown in Fig. 4). One can see damaged actin fibers from the surface down to 30 µm depth (first row). Dark layers from 20 to 30 µm depth are not presented since no stained structures were found in that volume. At around 40 µm, the images show intact cardiomyocytes with striated actin fibers (second row). At around 70 µm in depth, well-connected cardiomyocytes are visible (third row). The thickness and the orientation as well as the distance between the fibers were measurable, especially in the zoomed image by LiL-SIM. Despite the gain in resolution in all reconstructed planes down to 70 µm, aberration artifacts arise from 60 µm on due to phase distortion. Nevertheless, optical sectioned images were acquired from higher depths up to 120 µm, which equals the maximum working distance of the objective lens. For comparison, the third row includes three WiL-2PM images of selected planes taken at 5 µm, 40 µm and 60 µm depth. The insets (a) and (b) compare the resolution and contrast enhancement of the 5 µm deep image plane. In the profile plot below taken along the dashed line of inset (a) (LiL-SIM), we found three peaks with 312 nm, 345 nm and 177 nm FWHM (from left to right). According to the Gaussian fit curves, the distance of the adjacent peaks amounts to 146 nm. Insets (c) and (d) are taken at 55 µm depth comparing LiL-2PM and LiL-SIM. The profile plots below make it clear that although both modalities profit from the camera's LSS mode, LiL-SIM not only enhances the resolution but also the contrast. This proves again that the exposure band in LSS mode is not used for confocalization but only to reduce scattered light contributions from the same plane. Conclusively, by making use of the

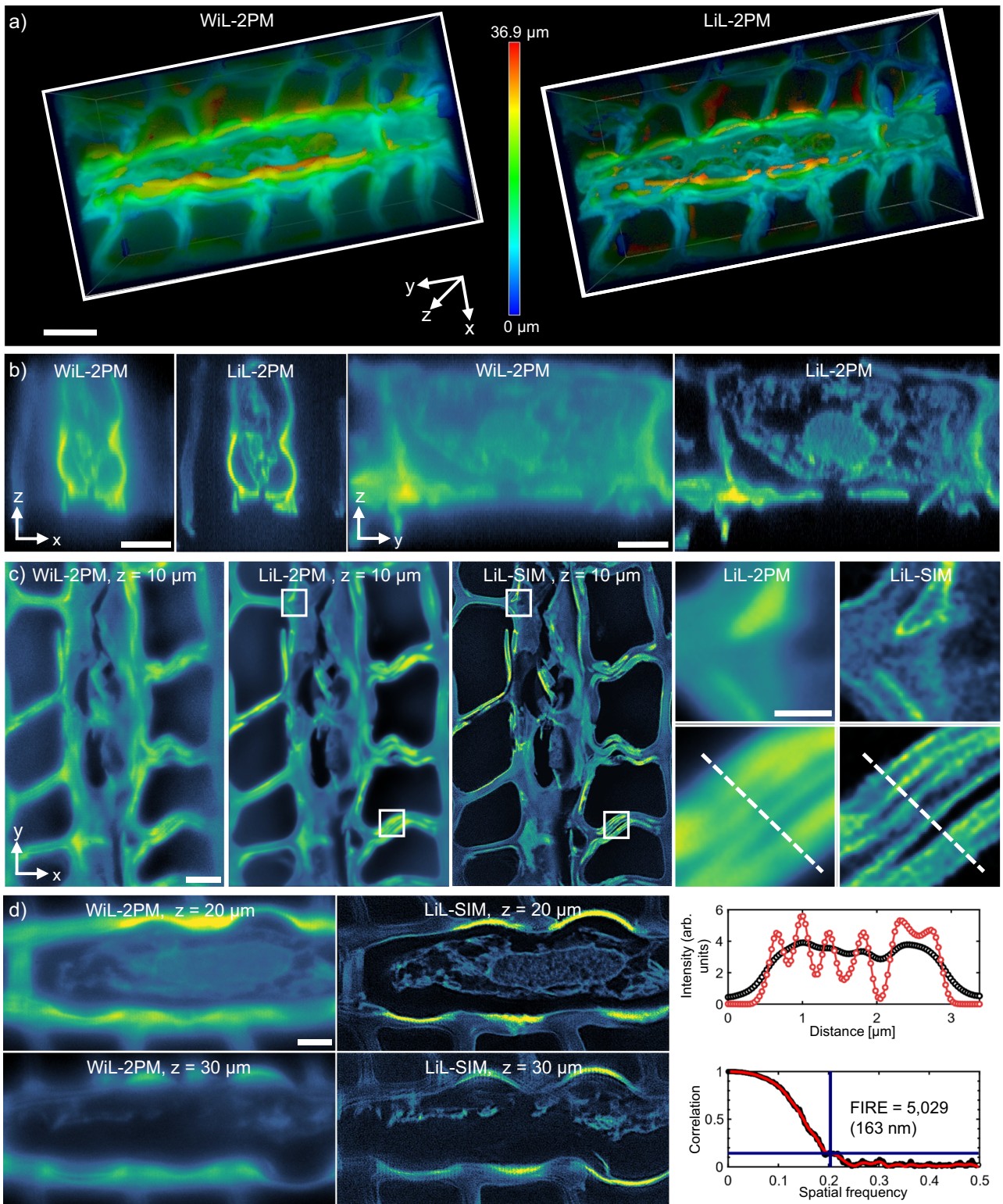

**Fig. 3 | Extending the penetration depth of super-resolved imaging in *Pinus radiata* tissue. a** Volume stacks acquired with WiL-2PM and LiL-2PM demonstrate the enhanced optical sectioning effect when using LSS mode. **b** Axial center cross-sections in xz and yz. **c** Comparison of WiL-2PM, LiL-2PM and LiL-SIM images of *Pinus radiata* at a pattern spacing of 300 nm and an imaging depth of 10 μm into the sample. Magnified LiL-SIM insets of the white dashed regions of interest indicate clear resolution improvement compared to the LiL-2PM insets. Line width comparison of LiL-2PM (black) and LiL-SIM reconstruction (red) along the dashed line shown in the insets. FRC data (black) and fit (red) indicate a resolution of 163 nm, taken at a correlation factor of 0.143 (blue bar). **d** WiL-2PM vs. LiL-SIM at 20 and 30 μm imaging depth. Resolution improvement and contrast enhancement shown in (**a**–**c**) has been verified in (*N* = 10) volume stacks acquired at distinct locations. The line profile is a representative curve out of (*N* = 5) individual measurements. FRC curve was taken out of (*N* = 5) measurements, evaluated for each presented imaging depth. Scale bars **a**, **b** 10 μm. **c** 5 μm, inset 1 μm. **d** 2 μm.

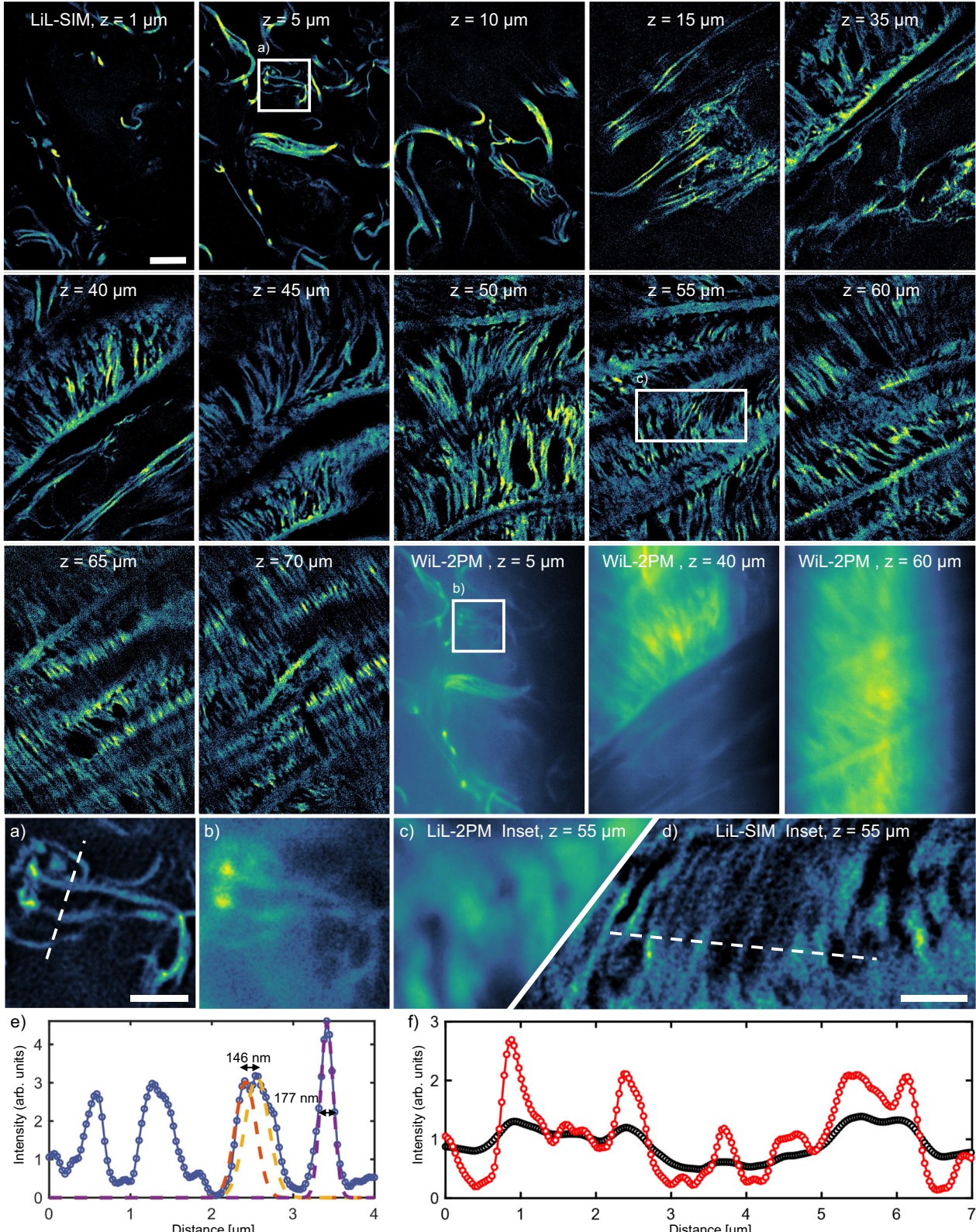

**Fig. 4 | LiL-SIM imaging in dense mouse heart muscle tissue.** Upper panel: Two-photon excited fluorescence images of mouse heart muscle acquired with LiL-SIM in varying depths from 1 to 70 μm. The depth at which the images were acquired is indicated at the top of each image. Up to an imaging depth of 50 μm, the pattern line spacing was set to 350 nm and to 400 nm for higher imaging depths. The LiL-SIM insets **a** at 5 μm and **d** at 55 μm show improved resolution compared to WiL-2PM and LiL-2PM insets (**b**, **c**). **e** Line profile (blue) taken along the dashed line in LiL-SIM inset (**a**). The distance between Gaussian fit 1 (orange) and Gaussian fit 2 (yellow) is 146 nm, while Gaussian fit 3 (purple) has a FWHM of 177 nm. **f** Comparison of line profiles from inset (,**c**, **d**) between LiL-SIM (red) and LiL-2PM (black). Data shown is a representative volume stack out of (N = 3) stacks taken at distinct locations. Line profiles have been evaluated in (N = 5) individual measurements for each presented imaging depth. Scale bar: 5 μm, left inset 1 μm, right inset 2 μm.

optical sectioning capability of two-photon excitation, our technique successfully demonstrates super-resolution imaging down to 70 μm penetration depth and 'confocal' imaging down to 120 μm depth for this particular biological specimen.

## Discussion

We present a method for super-resolved deep tissue imaging by combining SIM with line-scanning two-photon excitation. We overcome the severe problem of vanishing modulation contrast in highly scattering samples by utilizing the camera's lightsheet shutter (LSS) mode. In this way, we achieve sufficient contrast for SIM reconstruction even at depths around 70 μm in mouse heart muscle tissue.

First, we show that camera detection with LSS-mode (LiL-2PM) is comparable to common point detection in two-photon microscopy (PMT-2PM), whereas light scattering leads to blurred images using RS mode (WiL-2PM). The adaptation of LSS-mode to SIM is not straightforward but is accomplished by using off-the-shelf devices such as a cylindrical lens, a Dove-prism, a half-wave plate, and a rotation stage. The analysis of the zebrafish sample demonstrates the improved modulation contrast even at large depths and for line spacings close to the diffraction limit. Our insight is to not rely solely on the excitation modulation contrast to determine the alignment and functionality of SIM setups as commonly reported in literature[36,37]. In contrast, our focus on detection modulation led us to favor line-scanning over interference-based pattern generation. Although this switch has its disadvantages as discussed below, our experimental results have shown that enabling the camera's LSS mode significantly extends the penetration depth for super-resolution in dense samples.

Next, we quantified the super-resolving power with image decorrelation analysis and FWHM analysis, which are widespread methodologies among SIM microscopists: we determined the resolution improvement factor to be 1.94 by comparing the two imaging modalities LiL-SIM and LiL-2PM. Other groups published similar results for two-photon SIM: giving a few examples with bead samples, the resolution improvement was reported to be 2.14x[26], 1.97x[38] and 2.46x[39]. Studying a biological sample of U2OS cells, an improvement factor of 2.25x was reported in ref. 40. Further details on these literature values can be found in Supplementary Table T3. If, on the other hand, any optical aberrations and noise contributions introduced by the sample or the optical system are ignored, our experimentally determined factor is only 1.72 compared to the Rayleigh limit of the objective lens. While in cSIM this theoretical factor can reach 2.0, our simulations take the sequential pattern generation into account and derive 1.84 as an upper limit of our approach (see Supplementary Fig. S11). By imaging clusters of 190 nm sized beads, we were able to resolve individual beads, confirming that our gain in resolution is not due to edge enhancement or PSF deconvolution. The achieved super-resolution of $153 \pm 18$ nm (determined from 20 decorrelation measurements) was maintained in heart muscle tissue down to 70 μm depth. Since resolution gain depends on various factors, such as the phase stability or modulation contrast to noise ratio[41], the depth-dependent resolution gain may differ for other samples.

Naturally, LiL-SIM also has its limitations. The imaging speed of LiL-SIM for instance is inherently limited by the need to collect sufficient 2P signal (which is a limiting factor in all 2P-based super-resolution methods). This is particularly important when using fluorescent markers with low two-photon absorption cross-sections, which somewhat limits the choice of fluorophores. Our method is further limited by the flyback time of the galvo-scanner (which is needed since the LSS mode only runs from top to bottom). The most significant current limitation, however, is the time that it takes for the field rotator to move to a new position and settle there. We demonstrate the speed at which LiL-SIM can image samples by acquiring LiL-2PM image data for a single illumination angle. This is demonstrated in Supplementary Movie 2, where data was acquired with a line exposure time of 1 ms,

resulting in a frame rate of ~4.4 Hz for continued imaging. The imaging speed of LiL-SIM can be further improved across all illumination pattern angles by replacing the Dove prism with a galvanometer-based K-mirror[42]. Furthermore, the FOV can be increased by extending the excitation line profile using more powerful lasers. The line illumination could be made more even by using a flat-top instead of Gaussian profile of the laser line focus and the method could be further extended in terms of its information content by using multi-color labeled samples.

In conclusion, we successfully demonstrate that our approach extends the application of SIM to high penetration depths in dense samples by imaging *Pinus radiata*, heart muscle and zebrafish at depths up to 70 μm without loss of super-resolution. Our super-resolution method comes at a low technical cost, which makes it promising to reach a broad community of microscopists.

## Methods

### Two-photon fluorescence excitation LiL-SIM setup

The optical setup is schematically shown in Fig. 1. A custom-built tunable femtosecond laser ($\lambda_{em} = 1600–1680$ nm, $P_{out} = 1$ W, $f_{Rep} = 100$ MHz) followed by a second harmonic generation (SHG) module is used for two-photon fluorescence excitation at a wavelength of 800–840 nm. For all experiments, an excitation wavelength of $\lambda_{ex} = 800$ nm was used. The beam is expanded to 2 mm $1/e^2$ by a  -4× telescope composed of plano-convex lenses L1 (Thorlabs, $f = 40$ mm, LA1422-B-ML) and L2 (Thorlabs, $f = 175$ mm, LA1229-B). A cylindrical lens (Thorlabs, $f = 50$ mm, LJ1695L1-B) focuses the beam in the vertical direction while maintaining beam collimation in the horizontal direction. This results in a narrow line at the aperture of the 2D galvo-scanner (Thorlabs, GVS002). The line intensity profile is projected into the back focal plane of the objective lens by a scan telescope composed of scan- and tube lenses (Thorlabs, $f = 50$ mm, SL50-CLS2 and $f = 200$ mm, TTL200MP). In order to generate the excitation pattern in LiL-SIM, this laser line profile is scanned in defined scan steps across the sample. Thus, the pattern formation is not due to interference, but due to the step interval between consecutive line profiles. Since the line exposure times is rather long (2–5 ms) compared to the step time (~300 μs), we assume that on/off modulation of the laser during the scanning step could likely lead to an improvement in modulation contrast but would only complicate the synchronization. A field rotation unit, driven by a piezo-electric rotation mount (Thorlabs, ELL14), is located between the dichroic mirror and objective lens. The module is composed of a half-wave plate (Edmund Optics, 39–173) and a Dove prism (Thorlabs, PS992M-A). This arrangement rotates the intensity distribution by an optical angle that equals half the mechanical rotation angle of the module (rotated intensity profiles in the back focal plane (BFP) for (1) 0°, (2) 60° and (3) 120° are indicated in Fig. 1a). The HWP compensates the angle dependent polarization shift introduced by the Dove prism. Finally, an objective lens (Nikon 100x, 1.49 NA) generates a diffraction-limited line-shaped PSF in the sample. Translation stages (Thorlabs, MTS50/M-Z8 and MT1/M-Z8) are used for the lateral and axial positioning of the specimen. The line-intensity profile is scanned across the sample, leading to fluorescence emission over the entire FOV. The fluorescence signal is collected in epi-direction, and is back rotated by the rotation unit. After reflection by the dichroic mirror (AHF, F76-705) and propagation through a short pass filter (AHF, F75-680), an emission filter (AHF, F37-630) and an achromatic lens L3 (Thorlabs, $f = 200$ mm, AC254-200-A-ML), the signal is directed to a scientific complementary metal-oxide-semiconductor (sCMOS) camera (PCO, panda 4.2). This camera can either be run in the RS or LSS mode, which is explained in detail in section "Extending the Penetration Depth with Lightsheet Shutter Mode". The power measured in the BFP of the objective lens ranges from 20 mW to a maximum of 200 mW. It is chosen dependent on type of sample and penetration depth. The detection arrangement leads to a pixel size of 65 nm in object space, which amounts to a

maximum FOV of 67 μm × 67 μm (1024 pixels). Hardware components, such as the galvo-scanner, the rotation mount and sample translation stages are integrated into a custom-written Matlab program[43] and controlled via a PCIe DAQ card (National Instruments, PCIE-6351) including a breakout box (National Instruments, BNC2110). Hardware synchronization is achieved by a custom-written Matlab program. Image acquisition and camera settings including the lightsheet shutter mode settings are controlled by open-source software Micro-Manager[44]. The parameter "light sheet mode exposure lines" is set to 7 for all experiments, which is the number of rows that form the exposure band. This number is selected depending on the NA, as well as on the excitation wavelength of the laser source. Consequently, the exposure band has a length of 1024 pixels and a width of 7 pixels, which spans a FOV of 66.56 μm times 455 nm in object space (65 nm/px). By setting the exposure time per line to 5 ms, the "light sheet mode line time" is automatically set to 714.286 μs (5 ms/7) by Micro-Manager, which amounts to a total exposure time of $5\,ms + 1023 \times 0.714286\,ms = 735.71\,ms$. The trigger mode is set to an external hardware trigger for frame synchronization. Conventional two-photon fluorescence excited confocal point scanning is achieved by using a commercial two-photon setup (Miltenyi Biotec, TrimScope Matrix).

## Image processing

All image data was processed in Fiji (ImageJ 1.54f) and analyzed with custom-written Matlab software (Matlab version R022b). The acquired image sets were processed in the following way: (1) digital back rotation of image sets (2) normalization of image sets to ensure equal brightness in stacks with different rotation angles (3) SIM image reconstruction with open-source software fairSIM[13] (4) flat field correction based on the measured excitation matrix in a sample with homogeneous fluorophore concentration (see Supplementary Fig. S13). Digital back rotation was accomplished by a custom-written Python program (see Supplementary Section 2). As explained above, we generated illumination patterns with five phase shifts and three rotation angles and reconstructed SIM-images with fairSIM (ImageJ implementation, git build id: 584010c43 standard build). The OTF was approximated using a NA of 1.49 and emission wavelengths of the detected fluorescence ranging from 525 to 650 nm. OTF attenuation was enabled (with default parameters $a = 0.990$ and FWHM = 1.20). Parameter estimation was carried out by running individual phase estimates. Both the Wiener filter, as well as RL-deconvolution with 10–50 iterations were used for image reconstruction. RL-deconvolution worked better with our image sets because ringing artifacts that appeared when using the Wiener filter were better suppressed. The fairSIM modulation depth estimates typically produced values ranging from good to usable. When approaching high imaging depths >50 μm, the estimate was "weak modulation". For resolution estimation, the Fourier Ring correlation plugin (BIOP ImageJ implementation 1.0.2)[45] and decorrelation analysis (ImageJ implementation v1.1.8)[32] were used. All reported uncertainties of the lateral and axial resolution in the manuscript correspond to the standard deviation.

## Sample preparation

Mice were euthanized by cervical dislocation, and their hearts were removed via thoracotomy and rinsed with pre-chilled PBS (pH 7.4) to remove blood. The heart tissue was then fixed in 4% paraformaldehyde (PFA, prepared in PBS) at 4 °C overnight. After fixation, the tissue was washed three times with PBS (5 min each) and immersed in 30% sucrose (prepared in PBS) at 4 °C overnight, until it sank. The tissue was then embedded in optimal cutting temperature (OCT) compound, rapidly frozen at −80 °C, and stored at −80 °C once fully solidified. For sectioning, the OCT-embedded tissue block was equilibrated at −20 °C for 10–15 min before being sectioned to a thickness of 120 μm using a cryostat microtome. The sections were carefully mounted onto pre-coated

Poly-L-Lysine slides and air-dried at room temperature for 30 min to 1 h. Immunofluorescence staining involved washing the fixed tissue sections three times with PBS (5 min each), followed by permeabilization with 0.5% Triton X-100 (prepared in PBS) for 10 min at room temperature. The sections were then blocked with 5% bovine serum albumin (BSA) at room temperature for 60 min. Phalloidin staining for actin filament (#ab176757, Abcam) was performed for 20 min. Finally, the sections were mounted by applying Vectashield H-1000 mounting medium (refractive index of 1.45). The slides were stored at 4 °C and protected from light. Since only one male mouse was used, the study design did not consider sex as a factor in the analysis. All animal procedures were conducted in accordance with relevant ethical regulations and were approved by the "Regierung von Oberbayern" (Government of Bavaria, Germany) under license number "Organentnahme §4-Schunkert".

The zebrafish larvae were treated with two fluorescent stains to highlight specific cellular structures. Nuclei were stained with Hoechst (Catalog No. 62249, Thermo Fisher Scientific Inc.). For visualizing actin filaments, Phalloidin-Atto490LS (Catalog No. 14479, Sigma Aldrich) was used. The larvae were placed in a three-well slide (Cat. No. 475565) and embedded in Eukitt mounting medium (Cat. No. 03989, Sigma Aldrich) to ensure the preservation of their anatomy. The sample was then covered with a high-precision coverslip (Cat. No. DH22, A. Hartenstein GmbH), providing the optimal conditions required for super-resolution imaging.

Bead samples were prepared from a 10 μl bead solution (Bangslabs, fluorescent PS microspheres 0.19 μm dragon green) that has been diluted in 10 ml dH20 for a 1:1000 dilution. The solution has been sonicated for 3 min to avoid multilayer clustering. 10 μl of the diluted solution were pipetted on a coverslip and air dried for 2 h.

*Pinus radiata* samples are commercially available (Catalog No. 5986003, Bresser) and were prepared from autofluorescent tissue sections extracted from the basal region, cut in rings (diameter of 3 mm, thickness of 50 μm) and mounted on coverslips.

A fluorescence calibration slide (Argolight, Argo-SIM Slide) has been used for calibration of the microscope system, as well as for the line pair measurements presented in Fig. 1d and Fig. S4.

The fluorescent slide (Thorlabs, FSK2) used for the modulation contrast measurements (Figs. S7, S12) is a commercially available product and was mounted on a coverslip.

## Reporting summary

Further information on research design is available in the Nature Portfolio Reporting Summary linked to this article.

## Data availability

Detailed wiring diagrams, technical notes, as well as a parts list to rebuild the LiL-SIM setup are available in the supplemental document and in Github repository at https://doi.org/10.5281/zenodo.15031580. Raw data of the acquired volumes, raw SIM reconstruction data and the reconstructed SIM image set (which also includes the widefield and the deconvolved images) for Figs. 1–4 of the main manuscript is available at https://doi.org/10.5281/zenodo.15031504. Source data is provided with this paper. Data underlying the results presented in the supplemental document exceeds the capacity of the repository but is available upon request.

## Code availability

Relevant code is available in the Github repository at https://doi.org/10.5281/zenodo.15031580.

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

## Acknowledgements

We acknowledge the preparation of the zebrafish sample by Dr. Stefanie Kiderlen and writing the script for digital back rotation by Tobias Mulser. Further, we acknowledge Thorlabs, Inc. for providing the ELL14 CAD model, which was used in this study under a Creative Commons Attribution License. T.He. acknowledges funding by Bayerisches Staatsministerium für Wissenschaft und Kunst, H.2-F1116.MUE/54/2, FSP Angewandte Photonik. T.Hu. acknowledges support by the European Union's European Innovation Council (EIC) PATHFINDER Open Programme under grant agreement No. 101046928. Z.C. acknowledges the funding support of the German Research Foundation (DFG 510049865), Corona-Stiftung Nachwuchsforschungsgruppe (Junior Research Group Grant), the German Centre for Cardiovascular Research (DZHK) ("Förderkennzeichen", ID: 81X3600510) and Sonderforschungsbereich SFB TRR 267 (DFG, 403584255, project B05). Open Access funding enabled and organized by project DEAL.

## Author contributions

P.B. designed the system, performed experiments and analyzed image data together with T.K. M.L. and Z.C. prepared the heart muscle samples. The idea originated from the discussion between P.B. and T.He. T.He. and T.Hu. directed and supervised the project. All authors discussed the results and contributed to the manuscript.

## Funding

## Competing interests

The authors declare no competing interests.
