## [Transparent Peer Review file · Nature Communications]

Super-resolution upgrade for deep tissue imaging featuring simple implementation

Corresponding Author: Professor Thomas Hellerer

Version 0:

Reviewer comments:

Reviewer #1

(Remarks to the Author)

The manuscript "Super-Resolution Upgrade for Deep Tissue Imaging" presents a significant advancement in the field of deep tissue imaging, focusing on the development of a super-resolution microscopy technique that enhances the capabilities of two-photon laser-scanning microscopes for biological research. This work introduces an easy-to-implement method that substantially increases resolution using cost-effective optical modifications, making it an appealing solution for a broad range of researchers.

The authors propose a novel combination of two-photon excitation and structured illumination microscopy (SIM) with innovative use of a cylindrical lens, field rotator, and sCMOS camera. The use of the lightsheet shutter (LSS) mode in conjunction with patterned line-scanning to improve both depth penetration and resolution is particularly innovative. This combination results in up to a twofold resolution improvement, extending super-resolution imaging capabilities to depths of at least 70 μm in dense biological tissues, a substantial improvement over conventional methods. The manuscript stands out due to its original approach to overcoming the challenges of imaging in scattering tissues and its use of cost-effective solutions.

The presented method holds immense applicability for advanced bioimaging, particularly in studying complex tissues such as plant structures, muscle tissues, and zebrafish embryos. The flexibility of the system, which supports different fluorescent markers and objective lenses, makes it a highly versatile tool for labs working with diverse biological samples. Its potential to be integrated into existing two-photon microscopes with minimal modifications also enhances its practical appeal. The technique can benefit research areas such as developmental biology, neuroscience, and cardiology, where high-resolution imaging in thick tissues is essential.

The manuscript is well-written, clear, and easy to follow despite the technical complexity of the subject matter. The structure is logical, with clear sections explaining the motivation, methodology, results, and implications of the work. However, there are occasional instances where further clarification could enhance readability, particularly in the explanation of some of the more technical aspects (e.g., the role of field rotation and the use of LSS mode). A few grammatical and typographical errors should be addressed in a final proofreading.

Besides that, the manuscript provides comprehensive coverage of the method's development, including the underlying principles, technical details, and experimental results. The authors offer a detailed comparison of their method with existing techniques and include quantitative measurements of resolution improvements using both biological and synthetic samples. The inclusion of decorrelation analysis and Fourier ring correlation (FRC) to quantify resolution improvements adds rigor to the study. However, some additional discussion on the limitations of the method, particularly in terms of imaging speed and potential artifacts at greater depths, could provide a more balanced perspective.

In summary, this work has the potential to significantly impact the field of super-resolution microscopy, especially in deep tissue imaging. It presents a highly original and impactful contribution to super-resolution microscopy. Its innovative approach, broad applicability, and cost-effectiveness position it as a valuable tool for advancing bioimaging. With minor improvements in the clarity of some sections and additional discussion of limitations, the manuscript will be a strong candidate for publication in Nature Communications.

I have a few small recommendations for improving the manuscript:

- Address minor grammatical issues and improve clarity in the technical explanations.
- Consider expanding the discussion on the limitations of the method, particularly at higher depths.
- Include a brief comparison with other state-of-the-art techniques in terms of imaging speed and potential artifacts.

Reviewer #2

(Remarks to the Author)

In this manuscript, Byers and coworkers report a two-photon line-scanning structured illumination microscope (SIM) which improves lateral resolution at depths far from the coverslip. Other techniques have achieved similar performance using multiphoton microscopy, albeit not in the same manner as demonstrated here. The authors sync the line-scanning illumination with the light sheet scanning readout mode of an sCMOS camera to reduce out-of-focus fluorescence, thereby improving imaging contrast and SIM reconstruction. The authors call this SIM implementation Lightsheet Line-scanning SIM (LiL-SIM), and compare its performance to two other imaging modes, WiL-2PM (using the conventional rolling shutter mode of an sCMOS instead of light sheet readout mode) and LiL-2PM (acquisition as in LiL-SIM but superimposing the raw images rather than combining them via a Wiener filter). Judging from the images and data in the manuscript, we agree that LiL-SIM does have a higher lateral resolution than the other two modes. We commend the authors on the variety of (fixed) biological samples they image, including *Pinus radiata*, mouse myocardium and zebrafish, to demonstrate the application value of LiL-SIM. However, several aspects of the current manuscript dampen our enthusiasm for this paper:

1. The authors assert in their title and imply in their manuscript that their method provides an easy super-resolution 'upgrade' or 'implementation'. However, we see no evidence for this assertion, and in our experience implementing new SIM hardware or software is definitely not 'easy' to do properly. The authors only vaguely describe how their electronics are synchronized, making it hard for us to believe how adopting this method will be easy for other users.
2. Line 29, abstract mentions 'free choice of fluorescent markers' and 'microscope objective lenses'. Is this really true? It appears that all images were acquired with a single objective lens in this manuscript.
3. Lines 22, 30 in abstract and line 98 in introduction mentions 'inexpensive'. Is this detailed anywhere in the manuscript, and inexpensive relative to what?
4. Lines 45-46 in abstract mentions 'Localization microscopy is, however, only applicable to fixed samples because it is very time-consuming having to acquire thousands of images...'. Please amend or delete this claim, there are by now many studies that show the use of localization microscopy on live samples (for an early example, see <https://pubmed.ncbi.nlm.nih.gov/18408726/>).
5. Another route to improving imaging performance at depth is via multiview imaging. This has been combined with line-like SIM illumination along with a synchronized rolling shutter using 1p illumination, e.g. <https://pubmed.ncbi.nlm.nih.gov/34837071/>. This strategy and setup are more complex than described in this manuscript, but also uses a diffraction-limited line excitation in conjunction with the synchronized light sheet readout to suppress background/improve contrast, and thus also seems worth citing.
6. Lines 60-63, '... that the contrast can reach up to 100 percent ...'. Please provide citations and explain the definition of 'contrast' here.
7. After reading through '2. Results' and '4.1 Two-photon fluorescence excitation LiL-SIM setup', we still do not understand how the authors generate the stepwise line modulation patterns shown in Fig 2. f) in LiL-2PM and LiL-SIM and Fig. S2. Since the authors did not generate this grating-like line pattern by interference (using e.g., an SLM), and they appear to galvo-scan a single-line instead of scanning multiple illumination lines to generate this SIM pattern, we are mystified as to how the line is modulated and turned on/off. Is there a laser power regulation mechanism so that the femtosecond laser power is also temporally modulated within one image readout? This is important information that will be essential for those attempting to replicate the setup. Please explain in detail how the SIM illumination patterns are generated.
8. We could not understand the purpose of the half-wave plate (HWP) in the field rotator. As the authors claimed in lines 143-145: 'In this way, the contrast of the illumination pattern is not reduced by depolarization when the light is focused through a high numerical aperture (NA) objective lens.' However, the line-scanning SIM proposed in this manuscript is not based on interference. Therefore, we do not see the necessity to ensure s-polarization of excitation lasers as in coherent SIM (cSIM). Please provide citations to prove that the statement is correct and add figures to compare the differences in modulation difference with and without the HWP.
9. Aberrations due to field rotator are not mentioned or discussed. Presumably the glass in the dove prism introduces some aberration as light is being focused through, from air to glass and then back to air. Can the authors comment on this point? In particular, we are curious if there is any noticeable effect to how sharply a line can be focused relative to imaging without the Dove prism in place. This could be particularly relevant for the deviation of the experimental performance vs. the theoretical performance of the 'LiL-2PM' mode discussed in lines 222-223.
10. The authors claimed approximately 150-nm lateral resolution, using 190-nm diameter fluorescent beads. How is this possible? Please use beads with a diameter smaller than 150 nm and update Fig. 1c) accordingly.
11. Discussion of using line scanning in lines 150-157: '... , but also reduces the required laser power by a factor given by the number of lines that make up the final pattern (up to 200).' Reduce the laser power relative to what? The typical grating/SLM based SIM pattern? We do not understand the discussion here, please clarify for easier understanding.
12. Discussion of using different objectives lines 158-62: '...allowing the greatest possible flexibility in the choice of objective lenses...'. The authors also discuss the potential for using multiple objective lenses. Do they do this in practice?
13. Lines 176-178: 'We emphasize these points because switching from point detection to camera-based detection in two-photon microscopy degrades image quality, as discussed in subsection 2.3 and visualized in Fig. 2b.' But after comparing the two images shown in Fig 2 a) and b), we could not tell the difference between PMT-2PM and WiL-2PM from image quality or resolution. Please provide a more compelling comparison and quantify this statement for the images shown below

Fig. 2a), 2b).

14. Also in Figure 2, the authors compare their results to a standard 2PM (PMT-2PM), which we appreciate. However, they also claim in the caption that their technique (LiL-2PM) is superior in terms of S/B to PMT-2PM, which we find counterintuitive. Could they quantify this statement for the images shown below Fig. 2a), 2c)? Perhaps for a thin sample we could see how the combination of patterned excitation with confocal detection would result in better contrast than standard PMT-2PM, but it is also well acknowledged that for truly deep imaging in scattering tissue, camera-based detection is inferior to the PMT based detection in which all scattered illumination from a given excitation spot is assigned to that spot. Please discuss this issue and the 'crossover' point where PMT-2PM would be expected to outperform an imaging-based method like the authors.

15. In Figure 2 caption 'SIM images can be reconstructed with a modulation contrast higher than 0.1.' But we can still notice that there are some depths (e.g., 0-2 μm , 55-80 μm) where the modulation contrast is below the 0.1 threshold even in LiL-2PM mode. Then how to reconstruct images at these depths without artifacts?

16. In Fig. 3d, LiL-SIM @ $z = 30 \mu\text{m}$, we see some degree of line / honeycomb artifacts in the reconstruction, which look like residual patterned illumination. Please comment on this issue in the paper, in terms of its source and potential solutions.

17. Discussions, line 371, 'While in cSIM this theoretical factor can reach 2.0...' Is it truly true that for 2p imaging the expected factor is 2? Typically this assumes that excitation and emission wavelengths are approximately the same, which is not true here. The wavelength is 2x as long but there is an intensity squared effect, which would suggest an effective excitation PSF that is $\sqrt{2}$ smaller than using linear illumination at the same wavelength, but $\sqrt{2}$ larger than using 1p illumination at the usual 1p illumination wavelength. Therefore we are not sure if one would expect a full factor of 2.0 after SIM.

18. The authors mention 'low technical cost' or 'low cost' at multiple points throughout the manuscript without any real quantification of what these numbers mean. It is thus hard for us to evaluate this claim.

19. The authors appear to be using a 1.49 NA TIRF lens, which presumably is index-mismatched to the aqueous samples used here. Why was this lens chosen, it would seem to induce obvious spherical aberrations. Upon reading the methods section, it appears that all samples were mounted in Vectashield (RI 1.45), which we suspect is closer but not equivalent to the RI of the immersion oil. Please comment on this issue, and clarify in the main text how samples were mounted.

20. The number of exposure lines (corresponding to the confocal slit width) were not provided for images acquired using LiL-2PM and LiL-SIM, but this is an important experimental parameter, please report it.

Minor suggestions:

1. The combination of AO with multiphoton photon reassignment seems worth citing in the introduction, especially given its performance at depths greater than those reported here and with superior spatial resolution,

<https://pubmed.ncbi.nlm.nih.gov/28628128/>

2. Introduction, line 106: 'imaging depths well below 50 μm ' – Assuming the logical meaning is 'depth greater than 50 μm ' here, do the authors mean 'well above 50 μm '?

3. Please include abbreviations in the caption to Fig. 1 (e.g. define S, BFP, DM, F, L, TL, SL, SC, etc.). In this caption, 'suppressed' is spelled incorrectly as 'surpressed'.

4. The 'regular shutter (RS) mode' is unclear and is not a PCO-defined term. We suggest using 'rolling shutter (RS) mode' or the readout mode that the authors used specifically defined by PCO in their manual and SDK.

5. Methods: please provide complete vendor/model details for all parts, including the femtosecond laser and SHG module. If it is a home-built laser, please give a more detailed description.

6. SI section 2 on phase stability, line 20: 'homogeniousty' is not a word. 'homogeneity'?

7. Fig. S4B, graph axes (numbers and text) are annoyingly small and difficult to read.

8. SI section 5 discussion of pattern contrast, line 62, 'areal' -> 'area'

9. Please summarize all imaging parameters for figures in a table including imaging mode, sample / structure, laser intensity (W/cm^2), volume size, exposure time, total acquisition time etc.

10. SI section 6, 'vertical extend' -> 'vertical extent'

11. Video 1, we assume what seeing here is – on the left – 'WiL-2PM' and on the right, 'LiL-2PM'. Please clarify, either in the video caption or even better also on the video itself.

Reviewer #3

(Remarks to the Author)

Reviewer #4

(Remarks to the Author)

In this manuscript, the authors reported a new implementation of super-resolution structure illumination microscopy (SIM) with two-photon excitation for deep tissue imaging. The authors developed the imaging method with relatively low-cost optical components by taking advantage of sCMOS camera's lightsheet shutter mode. The authors compared the imaging capability of their new method, referred to as LiL-SIM, with conventional two-photon microscopy (referred to as PMT-2PM), and two additional intermediate imaging modalities enabled by their system (referred to as LiL-2P and WiL-2PM) in lateral resolution enhancement and deep tissue imaging qualities in *Pinus radiata*, mouse heart tissue, and Zebrafish larvae. The manuscript is innovative in the context of LiL-SIM optical implementation. However, I feel the overall improvement of imaging capabilities offered by LiL-SIM compared with other existing two-photon SIM in deep tissue imaging is marginal and I

recommend publishing in a more specialized journal. In any case, below are my detailed comments,

(1) In my opinion, the major tradeoff of the LiL-SIM system in achieving higher penetration depth is the relatively low imaging speed. In fact, the authors should clarify if the frame integration time (500 ms) is used to achieve the final LiL-SIM image or to capture one of the illumination patterns at a specific rotational angle. In other words, the authors should mention their imaging speed per frame in Hz for SIM imaging. Additionally, the manuscript would benefit from a discussion of the trade-offs between temporal resolution, spatial resolution, and image quality, providing guidance on which biological processes are suitable for imaging with the current setup.

(2) Following the above comment, I think the significance of the manuscript will be largely enhanced by imaging a live biological sample and performing some time-lapse LiL-SIM images.

(3) It would also be helpful to list down side-by-side comparisons between LiL-SIM and other state-of-the-art deep tissue high-/super-resolution imaging methods such as Adaptive optics-based SIM, conventional two-photon SIM, two-photon STED, and single-molecule localization-based methods on the key imaging capabilities, such as lateral and axial resolution, imaging speed, imaging penetration depth, depth of the field.

(4) The authors should report the axial resolution in LiL-SIM, and comment on the impact of using the striped two-photon illumination versus the point-scanning-based illumination on the axial resolution.

(5) Despite the use of cylindrical lenses reducing the optical complexity compared with adaptive optics, the benefit of low-cost and reduced complexity in LiL-SIM is not justified in the context of improving the scientific community or research. It would still require dedicated two-photon SIM training to implement the system. From a technology dissemination perspective, I suggest the authors to discuss and/or demonstrate how the low-cost and easy implementation features can directly help biological users who are interested in using two-photon SIM.

Reviewer #5

(Remarks to the Author)

Version 1:

Reviewer comments:

Reviewer #1

(Remarks to the Author)

The authors have satisfactorily answered all my comments. I think the presented method and implementation is an important addition to the toolbox of structured illumination microscopy, especially for deep-tissue imaging. I recommend publication as is.

Reviewer #2

(Remarks to the Author)

We greatly appreciate the revisions performed by the authors, and have only a few lingering minor comments that we would like to see addressed, please see attached.

Reviewer #3

(Remarks to the Author)

Reviewer #4

(Remarks to the Author)

The authors have partially addressed my concerns. However, since the content presented in the manuscript largely focuses on the accessibility and cost-efficiency other than pushing the technological limit. I still felt the manuscript does not meet the scope of Nature Communication considering the comparable imaging penetration, speed and resolution metrics achieved by similar methods. Also, considering the cost reduction, the outlined number of optical parts are around ~10.000-20.000 Euros, and I felt this level of reduction from ~30.000-40.000 Euros using EMCCD is relatively marginal. Therefore, I recommend considering publishing it in a more specialized journal such as Optica or BOE.

Reviewer #5

(Remarks to the Author)

We greatly appreciate the authors' efforts to improve their manuscript, which is indeed much improved. We have the final minor suggestions that we think should be addressed prior to publication:

-In response to reviewer 1's comment and line 491-492 in the revised manuscript 'By setting the exposure time for the entire frame to 5 ms, the "light sheet mode line time" is automatically set to 0.714286 ms (5 ms / 7) by Micro-Manager.' The description 'entire frame' is misleading because 5 ms should refer to the exposure time per line. The total exposure time of the entire frame is closer to $5 \text{ ms} + 1024 \times 0.714286 \text{ ms} = 736.43 \text{ ms}$. Please modify the description accordingly if our description is more accurate.

-The hardware timing diagram in SI is good addition (Fig S1), but we do not see the associated caption in the SI file:

Figure S1 a) Hardware diagram and b) timing diagram of the LiL-SIM setup.

-In response to our second comment about the choice of fluorescent markers, the authors claim in their rebuttal that they discuss these limitations in the Materials and Methods section of their manuscript. We did not see any discussion here, perhaps we missed it?

-We appreciate showing that the technique can be used with a 1.27 NA water lens, with the associated new supplementary figure:

-

Figure S5 WiL-2PM and b) LiL-SIM images of *Pinus radiata* when using a 60x / 1.27 instead of the 100x / 1.49 NA objective lens. c-d) Insets of the white dashed region shown in the main images. The double membrane wall with a distance of 210 nm can be resolved by LiL-SIM. Using the lower magnification objective lens increases the FOV. Scale bars: 20 μm , insets 5 μm .

It appears that the caption is missing 'a)' before 'WiL-2PM' in the first sentence of the caption. Also, it would be helpful to indicate by how much the FOV is increased, as mentioned in the caption, relative to the original 100x lens.

-We appreciate the additional clarity on the mechanism of stepwise line modulation. However, the Galvo voltage ramp shown in Figure S1 b) still may mislead the readers into thinking that the scanner is operating continuously rather than in discrete steps. Thus, we suggest adding a zoomed-in figure/inset to illustrate that the input voltage to the scanner is actually operated stepwise.

-We recommend additionally including the point mentioned in the authors' rebuttal, that modulating the intensity of the line during the scan would improve modulation contrast- perhaps a phrase to this effect could be included at the end of section 4.1, where the authors point out that modulation of the laser would complicate synchronization, but don't point out the improvement in contrast that is likely to be achieved by modulating the intensity of the laser.

-In response to our point 11, about reducing laser power, the comments in the rebuttal letter make sense. However, in the corresponding text in the manuscript, they appear to compare to 'widefield illumination required for 1-photon excited SIM...' We suspect that widefield 1-photon SIM requires far less power than applied here. Perhaps the authors should rephrase to something like, '...in comparison to the power required for full FOV 2p illumination, this also reduces the required laser power...'

-Fig2, we appreciate the changes and associated careful SNR/SBR analysis. Please specify in the caption/figure what the sample is in Fig. 2a-c, and indicate in the caption that the sample in Fig. 2a is distinct from that shown in Fig. 2b, c.

We thank the reviewers for their excellent feedback to our manuscript. Below, please find our detailed response to all the concerns and suggestions voiced by the reviewers. Also, we would like to note that we adapted the title to a slightly different wording by replacing “easy” with “simple”, but we are open for better suggestions by the reviewers.

Reviewer 1:

However, there are occasional instances where further clarification could enhance readability, particularly in the explanation of some of the more technical aspects (e.g., the role of field rotation and the use of LSS mode).

We thank reviewer 1 for this suggestion. To further clarify the technical specifications and aspects, a parts list has been added to the supplementary information. Further, timing diagrams for synchronization and the most essential code to generate the galvo voltage ramps have been added to the LiL-SIM Github repository (<https://github.com/Patby/LiLSIM>). In addition, we are pleased to further elaborate on the technical aspects of field rotation and the use of LSS mode:

We have added the following lines to section 2.3 in the manuscript: "Field rotation is critical for generating illumination patterns under different orientation angles, which finally leads to an isotropic resolution enhancement of LiL-SIM. Most commonly, SIM patterns are recorded at three different rotation angles of 0°, 60° and 120°. It is important to state that the mechanical rotation of the Dove prism by an angle α results in an optical field rotation of 2α . Consequently, if a field rotation of 60° is desired, the Dove prism needs to be rotated by 30°."

As already explained in the manuscript (section 2.3 – Field rotation), the fluorescence light is back rotated by passing the Dove prism in the backward direction, reverting the orientation change of the illumination pattern. Therefore, the pattern orientation on the camera remains static and is independent of the angular position of the Dove prism. The rotation accuracy of the piezoelectric rotation mount is specified with 0.05°, which allows for precise angular positioning.

Furthermore, the following lines were added to section 4.1 in the manuscript: "The lightsheet mode settings are adjusted internally by the control software Micro-Manager. The parameter “light sheet mode exposure lines” is set to 7 for all experiments, which is the number of rows that form the exposure band. This number is selected depending on the numerical aperture, as well as on the excitation wavelength of the laser source. Consequently, the exposure band has a length of 1024 pixels and a width of 7 pixels, which spans a field of view of 66.56 μm times 455 nm in object space (65 nm/px). By setting the exposure time for the entire frame to 5 ms, the “light sheet mode line time” is automatically set to 0.714286 ms (5 ms / 7) by Micro-Manager. The trigger mode is set to an external hardware trigger for frame synchronization."

Property Name	Preset Value
pco_camera-Exposure	5
pco_camera-Light Sheet Mode	On
pco_camera-Light Sheet Mode Exposure Lines	7
pco_camera-Light Sheet Mode Line Time	714.286
pco_camera-Triggermode	External

Figure 1: Lightsheet shutter mode configuration in MicroManager when using the PCO camera.

A few grammatical and typographical errors should be addressed in a final proofreading.

We thank the reviewer for careful proofreading and corrected grammar mistakes (highlighted in red) throughout our manuscript.

However, some additional discussion on the limitations of the method, particularly in terms of imaging speed and potential artifacts at greater depths, could provide a more balanced perspective.

We thank the reviewer for these suggestions. We have modified the Discussion and Conclusions section of the manuscript to discuss limitations of the method as follows: "Naturally, LiL-SIM also has its limitations. The imaging speed of LiL-SIM for instance is inherently limited by the need to collect sufficient 2P signal (which is a limiting factor in all 2P-based super-resolution methods). Our method is further limited by the flyback time of the galvo scanner (which is needed since the LSS mode only runs from top to bottom). The imaging speed of LiL-SIM can be improved by replacing the Dove prism with a galvanometer-based K-mirror [1]. Furthermore, the field of view can be increased by extending the excitation line profile using more powerful lasers. The line illumination could be made more even by using a flat-top instead of Gaussian profile of the laser line focus and the method could be further extended in terms of its information content by using multi-color labeled samples."

Regarding the potential artefacts, literature suggests that honeycomb patterns form due to decreased axial support in the acquired images [2]. For our particular technique, this can be caused by refractive index mismatches of the immersion medium, coverslip and specimen layers. This explains the formation of weak honeycomb patterns in the Pinus radiata depth measurement. Since the axial resolution decreases if those mismatches are present, this explains the honeycomb patterns in the Pinus radiata measurement (Fig. 3d, $z = 30 \mu\text{m}$). In response to the reviewer's comment, we have added the following line to the results section of this figure: "The weak honeycomb artefact patterns visible in Fig. 3d ($z = 30 \mu\text{m}$) are likely caused by refractive index mismatches of the immersion medium, coverslip and specimen layers resulting in a decrease of the axial resolution."

Reviewer 2:

Major Revisions:

1. The authors assert in their title and imply in their manuscript that their method provides an easy super-resolution ‘upgrade’ or ‘implementation’. However, we see no evidence for this assertion, and in our experience implementing new SIM hardware or software is definitely not ‘easy’ to do properly. The authors only vaguely describe how their electronics are synchronized, making it hard for us to believe how adopting this method will be easy for other users.

We thank the reviewers for this comment. Our assertion of the upgrade being "easy" was based on the assumption, that the LiL-SIM modality can be implemented on any two-photon (2P) excitation fluorescence microscope by adding the low-cost field rotation unit, an sCMOS camera, as well as some electronics and code required to synchronize the data acquisition process with the scanning process. We further explained the digital backrotation step, which allows for reconstruction of the raw image sets in open-source SIM reconstruction software. This is usually not common for advanced deep tissue super-resolution microscopy systems. We believe that in comparison to other 2P SIM techniques, our method neither requires extensive optical, nor extensive electronic modifications. Thus, most experienced 2P microscopists should be able to implement these modifications without huge efforts, which is a substantial improvement compared to state-of-the-art methods.

However, in order to address the reviewers' concerns, we replaced "easy" with "simple" in the title of the manuscript and added "cost-effective and relatively easy" to the abstract. Furthermore, we added a synchronization and timing diagram to the supplementary information file and provide the custom-written Matlab script to control the LiL-SIM microscope in our Github repository (<https://github.com/Patby/LiLSIM>). The implementation of LSS mode only requires the frame exposure start trigger signal to be directed to the camera. The camera settings, including line time, exposure time and exposure band width are controlled with open-source software Micro-Manager [2] and were set before the experiment. An additional trigger is needed for hardware triggering of the rotation mount. Only a frame trigger is needed to start the acquisition, which is available in every synchronized scanning microscope.

Figure 2: Hardware and timing diagrams of the proposed setup. a) The microscope is controlled via a custom-written MATLAB script named LiL-SIM GUI. Trigger signals and voltage ramps are generated and directed to the hardware components of the microscope via a data acquisition (DAQ) card, which is linked to a breakout box with physical BNC connectors. The camera settings including frame exposure time, line exposure time of the rolling shutter and lightsheet shutter mode settings are set by the open-source software Micro-Manager [2]. b) Timing diagram of the LiL-SIM setup. The camera is triggered via an external TTL pulse, which starts the frame acquisition. A flyback array is added at the end of each scan array to allow for smooth transition to the starting position. The individual voltage ramps are to enable phase shifting of the illumination pattern. After all corresponding images are recorded for a certain direction, ext. trigger 2 is applied to hardware trigger the rotation unit. This procedure is repeated until all images (usually 9 or 15 images) are acquired. The modification for hardware triggering the rotation mount specified in the components table can be found in the GitHub repository.

2. Line 29, abstract mentions ‘free choice of fluorescent markers’ and ‘microscope objective lenses’. Is this really true? It appears that all images were acquired with a single objective lens in this manuscript.

We would like to note that the method is, in principle, compatible with any choice of fluorescent markers and with any microscopy objective lens - but with the laser sources available to us, the choice of fluorophores is, of course, limited. We have modified the abstract by replacing "free choice" with "variety", which we believe is a more limiting phrase. The limitations with regard to the choice of fluorophores are then discussed further in the Materials and Methods section of the manuscript.

The use of multiple objective lenses is straightforward in LiL-SIM, since the laser line is centered in the back focal plane (independent of the rotation angle of the Dove prism). As long as the line extent covers the full BFP of the objective, and the tube lens is chosen correctly, it is easy to use various objectives. To our own surprise, we found that if the LSS mode is successfully calibrated for a certain objective lens, it doesn't even need to be recalibrated when swapping lenses. We demonstrate this by additional measurements using a 60x / 1.27 NA objective lens, which we added to the supplementary document (see Fig. R3, below). We further compared the resolution when using objective lenses with different magnifications in point 10.

Fig. R1: a) WiL-2PM and b) LiL-SIM images of *Pinus radiata* when using a 60x / 1.27 NA instead of the 100x / 1.49 NA objective lens. c-d) Insets of the white dashed region shown in the main images. The double membrane wall with a distance of 210 nm can be resolved by LiL-SIM. Using the lower magnification objective lens increases the FOV. Scale bars: 20 μm , insets 5 μm .

In response to this comment, we added Fig. R1 to the supplemental information file to demonstrate the use of different objective lenses with LiL-SIM.

3. Lines 22, 30 in abstract and line 98 in introduction mentions ‘inexpensive’. Is this detailed anywhere in the manuscript, and inexpensive relative to what?

We chose the word “inexpensive” in comparison to the overall cost of other 2P SIM implementations. We note, however, that this term is rather flexible and have also replaced it by “cost-effective”. The field rotator can be assembled for less than 1000 € by buying off-the-shelf components. We also use a low-cost sCMOS camera (< 10.000 €) and we were able to collect sufficient 2P signal for SIM reconstructions in depths of up to 70 μm with the LSS mode synchronization. Comparable state-of-the-art techniques use EMCCD cameras for the experiments, which are much more expensive (usually above 20.000 €) [3,4]. We added a parts list of the components including their purchase cost in the supplementary document as Table T4 and in our answer under “Minor Revisions, point 5”.

4. Lines 45-46 in abstract mentions ‘Localization microscopy is, however, only applicable to fixed samples because it is very time-consuming having to acquire thousands of images...’. Please amend or delete this claim, there are by now many studies that show the use of localization microscopy on live samples (for an early example, see <https://pubmed.ncbi.nlm.nih.gov/18408726/>).

The reviewers are correct with regard to this statement in the Introduction section of the previous manuscript and we apologize for making this rather broad claim. The limitation really only applies to methods that use organic fluorophores in combination with antibodies or anti-sense strands, such as e.g. dSTORM or DNA-PAINT, where samples also have to be permeabilized. In order to correct this, we decided to delete this statement altogether from the Introduction section.

5. Another route to improving imaging performance at depth is via multiview imaging. This has been combined with line-like SIM illumination along with a synchronized rolling shutter using 1p illumination, e.g. <https://pubmed.ncbi.nlm.nih.gov/34837071/>. This strategy and setup are more complex than described in this manuscript, but also uses a diffraction-

limited line excitation in conjunction with the synchronized light sheet readout to suppress background/improve contrast, and thus also seems worth citing.

We thank the reviewers for this comment. They are correct, and we have added a reference to the multiview imaging paper to our manuscript (Wu, Y., Han, X., Su, Y. *et al.*, *Nature* **600**, 279–284 (2021)).

6. Lines 60-63, ‘... that the contrast can reach up to 100 percent ...’. Please provide citations and explain the definition of ‘contrast’ here.

We refer to modulation contrast of the illumination pattern. This should state the difference between pattern generation based on interference (used in conventional SIM) vs. the incoherent pattern formation of point-scanning methods (i.e. ISM, instant SIM). The modulation contrast is evaluated by using the fairSIM algorithm. In response to this comment, we rephrased contrast to "modulation contrast" and added a reference (Heintzmann and Huser, *Chem. Rev.* **117**(23), 13890-13908 (2017)).

7. After reading through ‘2. Results’ and ‘4.1 Two-photon fluorescence excitation LiL-SIM setup’, we still do not understand how the authors generate the stepwise line modulation patterns shown in Fig 2. f) in LiL-2PM and LiL-SIM and Fig. S2. Since the authors did not generate this grating-like line pattern by interference (using e.g., an SLM), and they appear to galvo-scan a single-line instead of scanning multiple illumination lines to generate this SIM pattern, we are mystified as to how the line is modulated and turned on/off. Is there a laser power regulation mechanism so that the femtosecond laser power is also temporally modulated within one image readout? This is important information that will be essential for those attempting to replicate the setup. Please explain in detail how the SIM illumination patterns are generated.

We apologize for not explaining the image formation process in greater detail. In LiL-SIM a laser line is scanned in defined scan steps across the sample, and fluorescence emission is detected by a camera. The pattern formation is not originating due to interference, but due to the step interval between the individual scanned lines. Since the line exposure times are high (2-10 ms) compared to the step time (approx. 300 μ s), we found that modulation of the line during the scan step is not directly needed. However, doing so would definitely improve the modulation contrast.

In response to this comment, we have added the following text to section 4.1 of the manuscript: "In order to generate the excitation pattern in LiL-SIM, this laser line profile is scanned in defined scan steps across the sample. Thus, the pattern formation is not due to interference, but due to the step interval between consecutive line profiles. Since the line exposure times is rather long (2-10 ms) compared to the step time (approx. 300 μ s), we found that on/off modulation of the laser during the scanning step is not absolutely necessary but would only complicate the synchronization."

8. We could not understand the purpose of the half-wave plate (HWP) in the field rotator. As the authors claimed in lines 143-145: ‘In this way, the contrast of the illumination pattern is not reduced by depolarization when the light is focused through a high numerical aperture (NA) objective lens.’ However, the line-scanning SIM proposed in this manuscript is not

based on interference. Therefore, we do not see the necessity to ensure s-polarization of excitation lasers as in coherent SIM (cSIM). Please provide citations to prove that the statement is correct and add figures to compare the differences in modulation difference with and without the HWP.

To validate our claim, that the polarization of the line influences the pattern period, we removed the HWP from the rotation mount and inserted it in a second rotation mount in front of the galvo-scanner. We generated patterns with s- and p-polarization and compared the modulation depth (see Figure R4, below). There are many reports in the literature on the influence of polarization on the laser spot in the focal volume, which also relates to line-excitation. References to this can also be found in the pioneer work of Richards and Wolf[5,6].

In response to this comment, we have added the measurements shown in Figure R4 to the supplemental information file and also refer to this figure in the section "Field rotation" in the main manuscript.

Figure 3: a) Influence of polarization on the resulting illumination patterns. The modulation contrast in the p-polarized pattern is reduced due to depolarization by the high NA objective lens. b) Line plots extracted from both s- and p-polarized images quantify the increase in modulation contrast for s-polarized patterns. Scale bar c) 1 μm .

9. Aberrations due to field rotator are not mentioned or discussed. Presumably the glass in the dove prism introduces some aberration as light is being focused through, from air to glass and then back to air. Can the authors comment on this point? In particular, we are curious if there is any noticeable effect to how sharply a line can be focused relative to imaging without the Dove prism in place. This could be particularly relevant for the deviation of the experimental performance vs. the theoretical performance of the 'LiL-2PM' mode discussed in lines 222-223.

We thank the reviewers for bringing up this important point. We evaluated the field homogeneity by measuring an evenly distributed array of fluorescent rings (one of the test samples on the Argolight Argo-SIM slide) with our 100x objective lens (see Fig. R5a). No field distortion, nor spherical aberrations were introduced over the entire FOV (65 x 65 μm). We further carried out measurements with a 40x objective lens to cover the full extent of the array of rings. Fig. R5b) shows the measurement with the Dove prism inserted in the beam path, while we removed the Dove prism in Fig. R5c). We observed slightly decreased signal intensity when the Dove prism is inserted. This can be explained by pulse broadening due to dispersion. However, the full ROI remained homogeneous, and we could not find any evidence for additional aberrations or field distortion that were introduced by the prism over a FOV of 100 x 100 μm . It appears that the 200 mm focal length of the tube lens, which focuses the laser on the back focal plane of the objective,

only introduces such a small angle that these aberrations are negligible. The recorded fluorescence is not strongly influenced by the Dove prism, as the latter is placed in the "infinity beam path" of the microscope between the objective and the tube lens, which focuses it onto the camera for spatially resolved detection.

Figure 4: Evaluation of field distortion and aberrations when using the Dove-prism in the excitation and detection beam path. a) Recording the full FOV with the 100x objective lens with inserted Dove prism. b) Recording a FOV of 100 μm x 100 μm using the 40x objective lens (with Dove prism inserted). c) Recording of the same FOV by using the 40x objective lens without the Dove prism. No significant distortions or aberrations were detected. Rings are homogeneous over the entire FOV. Scale bars: a) 10 μm , b) 20 μm , inset 5 μm .

In response to this comment we have added the measurements shown in Figure R5 to the supplemental information file and also refer to this figure by adding the following lines to section 2.2 of the main manuscript: "Lastly, we also analyzed if the field rotator introduces aberration such as field distortion or if it negatively impacts the spatial resolution that can be achieved by this method. As detailed in Suppl. Fig. S8, we found that the field rotator has no negative impact on the imaging performance of the setup except for a slightly decreased signal intensity."

10. The authors claimed approximately 150-nm lateral resolution, using 190-nm diameter fluorescent beads. How is this possible? Please use beads with a diameter smaller than 150 nm and update Fig. 1c) accordingly.

We claimed 190 nm resolution on the bead sample with a well-defined dip in the corresponding line plot (Manuscript, Fig. 1c). This indicates that the lateral resolution might be higher. In the manuscript, we evaluate the resolution in the images based on Fourier-Ring correlation and decorrelation analysis. In order to fulfill the reviewer's request and for further quantification, we imaged fluorescent line pairs with decreasing distance ranging from 390 to 0 nm (shown in Fig. R6). This is one of the test patterns on the Argolight Argo-SIM slide. Fig R6a compares the line pairs of LiL-SIM and the deconvolved averaged image set when using the 100x/1.49 NA objective lens. It can be clearly seen that LiL-SIM improves the resolution significantly, which allows for resolving the individual lines down to a distance of 150 nm (shown in Fig. R6b). In Fig. R6c, the dip of the individual line pairs is shown for the 100x objective lens. Further, we repeated this evaluation for the 60x / 1.27 objective lens (Fig. 6d) and for the 40x / 1.15 objective lens (Fig. R6e). We are able to show that we can improve the resolution by using multiple objective lenses without any additional effort.

Figure 5: a) Comparison of resolvable line pairs (LP) by applying LiL-SIM vs Richardson-Lucy deconvolved LiL-2PM (10 iterations) imaging using a 100x / 1.49 NA objective lens. b) The 150 nm LP is resolved by LiL-SIM, while it cannot be resolved by deconvolved LiL-2PM. c) Line plots along the dashed line in the main image show the resolvable line pairs with the 100x objective lens. The spatial resolution is 270 nm for LiL-2PM and 150 nm for LiL-SIM. d) Corresponding line plot when using the 60x / 1.27 NA objective lens (LiL-2PM resolution: 360 nm, LiL-SIM: 210 nm). e) Corresponding line-plot when using the 40x / 1.15 NA objective lens (LiL-2PM resolution: 390 nm, LiL-SIM: 270 nm). Scale bars: a) 5 μm , b) 1 μm .

In response to the reviewer's comments, we added the line profile shown in Fig. R6c to Fig. 1 in the main manuscript. We also added the following lines to section 2.1 in the main manuscript: "In the final step, we determined the maximum resolution by measuring fluorescent line pairs (LPs) on a microscope calibration slide. Figure 1d shows a line profile acquired using the 100x / 1.49 NA objective lens, where the initial LP spacing is 390 nm, decreasing by 30 nm with each successive pair. This allows for determination of the lateral resolution of the proposed modalities: LiL-2PM resolves LPs down to 270 nm (black), while LiL-SIM achieves a resolution of 150 nm (red). Corresponding profiles for the 60x / 1.27 NA and 40x / 1.15 NA objective lenses are provided in the supplementary document." We also added the entire figure R6 as Fig. S4 to the supplemental information file.

11. Discussion of using line scanning in lines 150-157: ‘..., but also reduces the required laser power by a factor given by the number of lines that make up the final pattern (up to 200).’ Reduce the laser power relative to what? The typical grating/SLM based SIM pattern? We do not understand the discussion here, please clarify for easier understanding.

We apologize if this sentence has lead to confusion. We wanted to state that 2P widefield excitation is not possible, since it would require a substantially higher laser power in addition to other measures, e.g. temporal focusing. Instead of generating an illumination over the entire FOV, we narrow down the excitation volume to a line-intensity distribution. If 200 lines are used for generating the full illumination pattern, we can reduce the power of the laser source by a factor of 200, compared to a full FOV 2P illumination.

In response to the reviewer's comments, we modified the text in this section to hopefully alleviate the confusion: "This improves the detected modulation contrast and thus increases the penetration depth. Furthermore, in comparison to the widefield illumination required for 1-

photon excited SIM, this also reduces the required laser power by a factor given by the number of lines that make up the final pattern (up to 200)."

12. Discussion of using different objectives lines 158-62: ‘...allowing the greatest possible flexibility in the choice of objective lenses...’. The authors also discuss the potential for using multiple objective lenses. Do they do this in practice?

Please see our response and the additional information provided in response to point 2 (Fig. R1) and 10 (Fig. R6), above.

13. Lines 176-178: ‘We emphasize these points because switching from point detection to camera-based detection in two-photon microscopy degrades image quality, as discussed in subsection 2.3 and visualized in Fig. 2b.’ But after comparing the two images shown in Fig 2 a) and b), we could not tell the difference between PMT-2PM and WiL-2PM from image quality or resolution. Please provide a more compelling comparison and quantify this statement for the images shown below Fig. 2a), 2b).

We apologize for this shortcoming. In response we have acquired additional data to provide a clearer comparison. This data, also with a detailed explanation and quantification of signal-to-background ratio (SBR) and signal-to-noise ratio (SNR) can be found below in point 14.

14. Also in Figure 2, the authors compare their results to a standard 2PM (PMT-2PM), which we appreciate. However, they also claim in the caption that their technique (LiL-2PM) is superior in terms of S/B to PMT-2PM, which we find counterintuitive. Could they quantify this statement for the images shown below Fig. 2a), 2c)? Perhaps for a thin sample we could see how the combination of patterned excitation with confocal detection would result in better contrast than standard PMT-2PM, but it is also well acknowledged that for truly deep imaging in scattering tissue, camera-based detection is inferior to the PMT based detection in which all scattered illumination from a given excitation spot is assigned to that spot. Please discuss this issue and the ‘crossover’ point where PMT-2PM would be expected to outperform an imaging-based method like the authors.

Before we quantify the SBR and SNR, we wanted to determine and compare the resulting SBR when either using point-detectors or spatial detectors (i.e. cameras). The SBR mainly depends on two properties: (1) the lateral and axial extent of the PSF and (2) the scattering properties of the specimen. In the case of conventional spot-scanning 2P microscopy (including a point detector), the optical transfer function (OTF) of the system can be described as the autoconvolution of the 2P excitation OTF. Contrary, the system OTF of camera-based 2P microscopy is expressed as the convolution of the 2P excitation OTF and the 1P emission OTF. Consequently, the (axial) missing cone is more efficiently filled in the case of conventional point-scanning 2P microscopy, while camera-based 2P microscopy yields an increased lateral and axial extent of the system OTF, resulting in higher lateral and axial resolution. It is not straightforward to discuss a crossover point here, because the achieved optical sectioning fundamentally depends on the density and scattering properties of fluorescent structures contained in the specimen. Despite the fact, that the SBR of a point-detector based setup should always be increased compared to a camera-based setup, camera detection is mandatory in order to reconstruct SIM images.

Figure R7. Impact on signal-to-background ratio (SBR) and signal-to-noise ratio (SNR) for different imaging modalities. (a–c) 3D volume renderings of the same sample acquired with a) point-scanning setup with a photomultiplier tube detector (PMT-2PM), b) line-scanning with a camera detector using the rolling shutter mode (WiL-2PM), and c) line-scanning with lightsheet shutter mode (LiL-2PM). Please note that a) was acquired with an entirely different setup than b) and c). (d–f) Single-plane images showing vessel structures at an imaging depth of approximately 20 μm below the surface. (g) Signal-to-background ratio (SBR) plotted as a function of imaging depth for the three configurations. The results demonstrate that PMT-2PM achieves higher SBR at greater depths than LiL-2PM and WiL-2PM. (h) Signal-to-noise ratio (SNR) as a function of imaging depth. The signal-to-noise ratio is increased for camera-based detection due to the longer integration times per pixel. Scale bars a–c) lateral and axial 10 μm, d–f) 10 μm.

We used a commercially available multi-photon setup (LaVision TriM Scope™ Matrix multi-photon microscope) and set similar frame times and excitation wavelengths for the acquisition to allow for a comparison to the camera-based LiL-SIM system. However, the laser power is substantially lower when using point-excitation (i.e. 7 mW compared to 200 mW when using the line-scanning setup). Fig. R7a compares the imaging modalities of a) point-scanning two-photon microscopy (PMT-2PM), where a photomultiplier tube is used as detector. Volume stacks of the same size, acquired in similar regions with WiL-2PM and LiL-2PM are shown in Fig. R7b–c, respectively. In Fig. R7d–f single planes with resin ducts, including xylem vessels are depicted in planes approximately 20 μm below the surface. We quantify SBR and SNR by evaluating the intensity inside xylem vessels (background) vs. the intensity of the vessel membrane (signal). This is achieved by measuring the signal intensity along the ring. Further, the background signal is evaluated by taking the mean of a 20x20 pixel area inside vessels. Ten vessels were evaluated per

imaging depth and the SBR was evaluated starting at an imaging depth of 5 μm . SBR and SNR curves comparing the modalities are shown in Fig. R8g-h.

As the reviewer stated correctly, the SBR of PMT-2PM is increased when using the commercial setup and decays for higher imaging depths. However, since the individual pixel dwell times are set substantially longer in a camera detector (1.16 μs vs. $\geq 5\text{ms}$ / pixel), the SNR is severely increased when using the camera modalities.

In response to these comments, we added to the supplementary information document a new section "4 SBR / SNR analysis" including the above Fig. R7 as Fig. S3. Furthermore, we have replaced in the main manuscript the less convincing images of the second row in Fig. 2 with the subset d), e) and f) from this figure R7. We added in section 2.3 : "The evaluation of SBR and SNR were calculated by dividing the mean of a signal area by the mean of the background, and by the signal standard deviation, respectively (further described in Suppl. Fig. S3). While PMT-2PM has the highest SBR, LiL-2PM offers an improvement of almost a factor of 2 compared to WiL-2PM. In terms of SNR, WiL-2PM and LiL-2PM have significantly increased SNR because of longer integration times per pixel (10 ms for WiL- and LiL-2PM vs. 1.16 μs for PMT-2PM)."

15. In Figure 2 caption 'SIM images can be reconstructed with a modulation contrast higher than 0.1.' But we can still notice that there are some depths (e.g., 0-2 μm , 55-80 μm) where the modulation contrast is below the 0.1 threshold even in LiL-2PM mode. Then how to reconstruct images at these depths without artifacts?

The stack presented in Fig. 2 starts at the coverslip position ($z = 0 \mu\text{m}$). The first structures appear at a depth of $z = 2 \mu\text{m}$. At imaging depths of $z = 55 \mu\text{m}$ and higher, the density of the specimen permits reconstruction without severe artefacts. However, this depends on the optical properties of the structure gradients in the focal volume and the density of the sample (from $z = 0$ up to the imaging position). Therefore, the maximum imaging depth of SIM based on the modulation contrast cannot be quantified in an absolute manner. To estimate the maximum imaging depth, we imaged various ROIs in zebrafish and heart muscle samples and were able to reconstruct SIM images up to 55 μm deep in zebrafish tissue and up to 70 μm deep in heart muscle tissue. In the heart muscle measurement presented in Fig. 4, the modulation contrast values were low (approx. 0.10-0.12) for the first imaging planes up to $\sim 10 \mu\text{m}$ and was found to be increased (approx. 0.15-0.16) for the planes between 40 and 60 μm . Hence, the modulation contrast not only depends on the imaging depth, but also on the presence of structures resulting in a signal modulation. However, the modulation contrast gradually decreases for higher imaging depths down to a value of 0.11 and 0.10 for imaging planes at 65 and 70 μm , respectively.

16. In Fig. 3d, LiL-SIM @ $z = 30 \mu\text{m}$, we see some degree of line / honeycomb artifacts in the reconstruction, which look like residual patterned illumination. Please comment on this issue in the paper, in terms of its source and potential solutions.

Such artifacts are commonly observed in widefield SIM microscopy based on missing axial frequency support [8]. Simulations also showed that low SNR levels can lead to residual background. Especially at high imaging depths, signal levels are substantially lower, and the refractive index mismatch extends the PSF along the axial direction. Therefore, the axial support decreases even if 2P microscopy and LSS mode are used, which can lead to the observed reconstruction artifacts. In conclusion, the lowered axial resolution results in the lower modulation contrast value at $z = 30 \mu\text{m}$.

As mentioned in our response to reviewer 1 we added the following line to the results section of this figure: "The weak honeycomb artifact patterns visible in Fig. 3d ($z = 30 \mu\text{m}$) are likely caused by refractive index mismatches of the immersion medium, coverslip and specimen layers resulting in a decrease of the axial resolution."

17. Discussions, line 371, 'While in cSIM this theoretical factor can reach 2.0...' Is it truly true that for 2p imaging the expected factor is 2? Typically this assumes that excitation and emission wavelengths are approximately the same, which is not true here. The wavelength is 2x as long but there is an intensity squared effect, which would suggest an effective excitation PSF that is $\sqrt{2}$ smaller than using linear illumination at the same wavelength, but $\sqrt{2}$ larger than using 1p illumination at the usual 1p illumination wavelength. Therefore we are not sure if one would expect a full factor of 2.0 after SIM.

We apologize if this sentence has caused confusion. We obviously refer to the theoretical factor of 2 in 1P excited coherent SIM, since interference-based approaches with 2P microscopy have not been reported to the best of our knowledge. It is true that the 2P excitation PSF is $\sqrt{2}$ larger than the 1P excitation PSF at half the 2P excitation wavelength. However, most 2P excitation spectra are substantially blue shifted (i.e. the peak wavelength of the 2P absorption spectrum is lower compared to double the peak wavelength of the 1P-absorption spectrum). In our experiments, we excite Alexa Fluor 594 with a 2P excitation wavelength of 800 nm (compared to the peak absorption wavelength of 581 nm using 1P excitation). PSF simulations show that the FWHM values are almost equal when comparing both PSFs. Since the pattern spacing ultimately determines the resolution (SNR - SBR is not considered), 1P and 2P excitation should yield the same resolution improvement factor with these excitation sources.

18. The authors mention 'low technical cost' or 'low cost' at multiple points throughout the manuscript without any real quantification of what these numbers mean. It is thus hard for us to evaluate this claim.

Please see our response to point 3, above. We added a parts list of the components including their purchase cost in the supplementary document as Table T4 and in our answer under "Minor Revisions, point 5", below.

19. The authors appear to be using a 1.49 NA TIRF lens, which presumably is index-mismatched to the aqueous samples used here. Why was this lens chosen, it would seem to induce obvious spherical aberrations. Upon reading the methods section, it appears that all samples were mounted in Vectashield (RI 1.45), which we suspect is closer but not equivalent to the RI of the immersion oil. Please comment on this issue, and clarify in the main text how samples were mounted.

It is true that there is a refractive index mismatch between the layers, which leads to an axial increase of the PSF dependent on the imaging depth. Usually, water immersion objective lenses are preferred. However, the samples were provided by the Heart Center Munich with this particular mounting medium. These samples were prepared for the Leica microscope available at the Heart Center. To produce a sample with a different mounting medium would have required

us to sacrifice a mouse just for this purpose, which is, unfortunately, ethically problematic in Germany. Therefore, we decided to use an oil immersion lens with a “comparable” refractive index. Please note that the mounting medium Vectashield was used primarily because it is a soft setting mounting medium, which does not adversely affect the sample during the curing process compared to hard setting mounting media with different refractive index. We added information about the mounting medium to the Materials and Methods section in the main document.

20. The number of exposure lines (corresponding to the confocal slit width) were not provided for images acquired using LiL-2PM and LiL-SIM, but this is an important experimental parameter, please report it.

We apologize for this missing information. The number of exposure lines contained in the exposure band was set to 7 for these experiments. We added this information to the supplementary information.

Minor Revisions:

1. Included the AO reference in our manuscript.
2. Corrected to “well above 50 μm ”.
3. Abbreviations to Fig. 1 included.
4. We replaced regular shutter with rolling shutter.

5. Methods: please provide complete vendor/model details for all parts, including the femtosecond laser and SHG module. If it is a home-built laser, please give a more detailed description.

Components used to build the microscope are briefly collected in the parts list. Models marked in bold need to be additionally purchased to implement the LiL-SIM method into a conventional 2P microscope. This amounts to a total of below 10.000 € for the modification (including the sCMOS camera). We added this parts list in the Supplementary document (T4). Furthermore, we added a reference to the laser source in the main manuscript.

Part	Model/Vendor	Cost
Optical setup		
Laser	Toptica prototype	-
Cylindrical lens	LJ1695RM-B / Thorlabs	109,05 €
Scanner	GVS002 / Thorlabs	2.134,28 €
Scan lens	SL50-CLS2 / Thorlabs	3.250,27 €
Scanning tube lens	TTL200MP / Thorlabs	1.382,43 €
Piezoelectric rotation mount	ELL14K / Thorlabs	545,76 €
Half-wave plate	WPHSM05-808 / Thorlabs	461,09 €
Dove prism	PS992M-A / Thorlabs	241,88 €
Objective lens	CFI Apochromat TIRF 100XC Oil / Nikon	-
Dichroic	F76-705 / AHF	approx. 600 €
Cleanup filter	F75-680 / AHF	-
Detection filter 1	F37-630 / AHF	-
Detection filter 2	F37-584 / AHF	approx. 400 €
Detection tube lens	AC254-200-A-ML / Thorlabs	105,83 €

Camera	Panda 4.2 / PCO	approx. 8000 €
Hardware		
DAQ Card	PCIe 6341 / NI	approx. 1300 €
Breakout box	BNC2110 / NI	approx. 700 €
Z-stage	MT1/M-Z8, KDC101 / Thorlabs	approx. 1000 €
XY-stages	KMTS50E/M / Thorlabs	1.687,34 €
XYZ-hub controller	KCH301 / Thorlabs	approx. 600 €

6. Corrected to “homogeneity”.

7. Increased axis fonts.

8. Corrected to “area”.

9. Please summarize all imaging parameters for figures in a table including imaging mode, sample / structure, laser intensity (W/cm²), volume size, exposure time, total acquisition time etc.

The power in the BFP amounted to 200 mW which results in a peak intensity of 106 GW/cm² for all experiments. λ_{em} [nm] specifies the fluorophore emission wavelength. d_p [nm] specifies the pattern spacing. EBW = exposure band width in pixels. z_{Step} = step interval of the z-stacks. z_{range} = depth of the acquired volume (over a ROI of 1024x1024 pixels). t_{exp} = exposure time per line. t_{acq} = acquisition time for a single frame. We added this table to the supplementary document.

LiL-SIM setup:

Sample	Figure	λ_{em} [nm]	d_p [nm], EBW	z_{Step}/z_{range} [μ m]	t_{exp}/t_{Acq} [ms]
Manuscript					
Fluorescent beads	1	525	350 / 7	-	10/1280
Pinus radiata	2b, 2c	650	200 / 7	0.5/0-40	10/2230
Zebrafish	2d, 2f	550	350 / 7	0.5/0-80	10/1280
Pinus radiata	3	650	300 / 7	0.5/0-40	10/1490
Heart muscle	4	650	350-400 / 7	0.5/0-70	10/1280-1120
Supplementary					
Pinus radiata	S2, S5, S6, S9, S10	650	350 / 7	0.2 / 0 - 40	10 / 1280
Pinus radiata	S3, Video1	650	200 / 7	0.2 / 40	5 / 1115
Pinus radiata	Video2	650	350 / 7	-	1 / 76
Fluorescent slide	S7, S12	525	350, 400 / 7	0.5 / 0-120	5 / 640, 560
Argolight SIM slide	S4, S8	525	350 / 7	Var.	10 / 1280.

PMT-2PM setup:

Sample	Figure	λ_{ex} [nm]	Power [mW]	z_{Step}/z_{range} [μ m]	t_{Acq} [ms]
Manuscript					
Pinus radiata	2a	800	350 / 7	0.2/0-40	2100

10. Corrected to “extent”.

11. Added captions for Video 1.

Reviewer 3:

(1) In my opinion, the major tradeoff of the LiL-SIM system in achieving higher penetration depth is the relatively low imaging speed. In fact, the authors should clarify if the frame integration time (500 ms) is used to achieve the final LiL-SIM image or to capture one of the illumination patterns at a specific rotational angle. In other words, the authors should mention their imaging speed per frame in Hz for SIM imaging. Additionally, the manuscript would benefit from a discussion of the trade-offs between temporal resolution, spatial resolution, and image quality, providing guidance on which biological processes are suitable for imaging with the current setup.

We thank the reviewer for pointing out the importance of imaging speed for deep tissue applications. First, please allow us to clarify that our main goal was to develop an easy-to-implement, cost-effective method which doesn't require extensive optical modifications to an existing two-photon microscope compared to other, recently reported two-photon super-resolution methods with comparable capabilities. The temporal resolution in 2P microscopy is inherently limited by the need to collect sufficient signal to obtain high signal-to-noise ratios in the acquired images (which is the limiting factor in every 2P-based microscopy method). In our case, the time required to collect a single individual raw image frame is 500-1000 ms (depending on the signal strength, but this frame rate is considered fast for acquiring two-photon images with high SNR).

For calculating the temporal resolution, the following aspects need to be considered: **(1)** The size of the region-of-interest to be imaged, **(2)** the line exposure time (2-5 ms), **(3)** the flyback time of the galvo-scanner (min. 19 ms) and, finally, **(4)** the rotation time of the Dove prism rotation unit (min. 70 ms). This amounts to a total of approx. 8 s for the acquisition of all the raw image frames required for the reconstruction of a full super-resolved LiL-SIM frame (3 pattern phase shifts per orientation). This still represents a substantial gap to the instant 2P-SIM method [4], which allows for acquisition times of approx. 2 s per super-resolved frame after averaging. However, an EMCCD camera with a higher sensitivity was used compared to the "low-cost" sCMOS camera used in our experiments. We would also like to note that temporal averaging of the image data is, however, typically not needed in LiL-SIM since the same ROI is readily recorded multiple times with different rotation angles, resulting in an overall SNR increase of $\sqrt{3}$ of the final reconstructed images. Therefore, the requirement for having to record multiple images at different orientation angles is not really a disadvantage compared to other 2P-SIM approaches, because in these, temporal averaging is typically required to obtain sufficient SNR for high quality images.

As our technique offers the capability to push the speed to higher frame rates, we are currently mainly limited by the time that it takes for the rotation mount used in the field rotator to come to a complete stop after rotation. A 60° field rotation can be achieved in just 70 ms, which would increase the image acquisition time by 140 ms for two rotation steps. It does, however, take the rotation mount approximately 800 ms to fully settle after the rotation. This is the main limitation to the acquisition speed with our current setup.

Consequently, a further increase in temporal resolution could possibly be achieved by implementing either one of the following steps: (1) by using fluorophores with higher quantum yield, (2) by using detectors with higher sensitivity, (3) by increasing the average laser power, (4) by implementing custom acceleration and deceleration drive patterns for the rotation mount, or (5) by replacing the piezoelectric rotation mount with a galvanometer-based k-mirror, as stated in the Discussion section in the main manuscript.

In general, LiL-2PM offers an increased temporal resolution compared to point-detection based 2P microscopy (PMT-2PM), especially when using high power laser sources. Conventional femtosecond lasers usually offer average powers of at least 200 mW, which typically cannot be used in conventional laser-scanning 2P microscopy due to the resulting high peak intensities, which exceed the damage threshold of biological materials in the focal volume. Regarding all the influencing factors of temporal resolution, an evaluation of imaging speed cannot easily be made because all factors need to be considered. However, we can evaluate the imaging speed by comparing a commercially available 2P setup and our LiL-SIM setup as we demonstrated in Fig. R7, above. Here, the frame acquisition time of the point-based 2P image was adjusted such that it enables a fair comparison between the methods. If the same acquisition times are used in both systems, then the SNR in the LiL-2PM image is significantly better because of the substantially higher pixel dwell times (10 ms for LiL-2PM vs 1.16 μ s for PMT-2PM). This highlights the strength of camera-based detection compared to point-scanning 2P systems (including the fact that spatially resolved detection is needed for SIM-based methods).

(2) Following the above comment, I think the significance of the manuscript will be largely enhanced by imaging a live biological sample and performing some time-lapse LiL-SIM images.

We thank the reviewer for this suggestion, but do, unfortunately, not have access to a BSL-1 biosafety lab, which would allow us to conduct experiments with living organisms in the labs of the Physics Department according to the biosafety laws governing such work in Germany. However, in order to appropriately respond to this comment and to demonstrate the temporal resolution of LiL-SIM, we lowered the line integration time to 1 ms and performed SIM reconstructions along a single orientation on the *Pinus Radiata* sample (please see the new supplemental video 2). The integration time for a single frame is set to 76 ms (including the flyback time), covering an area of 512 x 512 pixels. If three phases are recorded, the acquisition time is now 228 ms for a single orientation, or 4,4 Hz.

In response to this comment, we have also added two supplemental videos to the manuscript. In supplemental video 1 we provide a side-by-side comparison of the improved background rejection and contrast of LiL-2PM compared to WiL-2PM with increasing imaging depth. We also refer to the new supplemental video 2 in the Discussion and Conclusions section as follows:

"The most significant current limitation, however, is the time that it takes for the field rotator to move to a new position and settle there. We demonstrate the speed at which LiL-SIM can image samples by acquiring LiL-2PM image data for a single illumination angle. This is demonstrated in Suppl. Video 2, where data were acquired with a line exposure time of 1 ms, resulting in a frame rate of approx. 4.4 Hz for continued imaging."

(3) It would also be helpful to list down side-by-side comparisons between LiL-SIM and other state-of-the-art deep tissue high-/super-resolution imaging methods such as Adaptive optics-based SIM, conventional two-photon SIM, two-photon STED, and single-molecule localization-based methods on the key imaging capabilities, such as lateral and axial resolution, imaging speed, imaging penetration depth, depth of the field.

We would like to briefly elaborate on the side-by-side comparison between the mentioned techniques. The lateral resolution of single-molecule localization microscopy (SMLM) is typically superior compared to deterministic methods such as SIM and STED. It is, however, not only the spatial resolution that can be achieved which is important, but the penetration depth to which

this resolution can be maintained is also very important for biological applications. In Fig. R8b, below, we compare the imaging depth of SRM methods based on their penetration capability. As shown in this figure, the penetration depth of any imaging modality can be substantially increased if 2P fluorescence excitation (red bars) is used instead of 1P excitation (black bars). Starting with the methods using 1P excitation, 1P confocal microscopy is currently the only method which can achieve imaging depths of about 100 μm [9] in native samples (i.e. samples that are not fixed and optically cleared). 1P SIM is mostly restricted to around 15 μm depth [10] because of the degradation of the modulation contrast in deeper sample planes. 1P STED is also limited to around 15 μm [11] due to the distortions of the STED beam caused by aberrations and scattering. However, implementations of adaptive optics can further increase the imaging depth of 1P SIM up to values of 50 μm [12]. Most localization-based methods are basically restricted to imaging at the coverslip surface. Some approaches to deep imaging using 3D-STORM and 3D-SOFI have been made, but the penetration depth doesn't exceed approx. 6 μm [13–16]. Vaziri et al demonstrated localization microscopy at depths of up to 10 μm by using two-photon temporal focusing for fluorescence excitation [17]. More recently, the group of Ralf Jungmann demonstrated imaging of cells with DNA-PAINT in combination with spinning disk confocal microscopy of entire cells in depths of up to 10 μm [18]

Also, recently two-photon-excited STED has been combined with adaptive optics (AO) and was demonstrated at imaging depths of up to 90 μm [19,20]. The implementations of 2P SIM accomplished imaging depths of up to 110 μm [3,4] even without the need for AO. For further information on the temporal speed of these methods, we refer to excellent reviews [21–23].

Figure 6: a) Lateral and axial resolution of super-resolution methods. b) Penetration depth of 1P (black bars) and 2P (red bars) implementations.

In response to this comment, we added figure R8 to the supplemental information file together with a brief explanation of the figure.

(4) The authors should report the axial resolution in LiL-SIM, and comment on the impact of using the striped two-photon illumination versus the point-scanning-based illumination on the axial resolution.

We thank the reviewer for bringing up the question about axial resolution. The axial resolution of line-scanning and spot-scanning laser microscopy are equal (both methods overfill the back aperture of the objective lens with a collimated beam, which mainly describes the achieved resolution) and is on the order of the wavelength of the excitation laser. However, the detector modality has an influence on both lateral as well as axial resolution. In PMT-2PM, the system PSF can be described by squaring the excitation PSF. However, when switching to camera detection, the system PSF is expressed as the product of the excitation PSF and the emission PSF (which depends on the emission wavelength of the fluorophores). Consequently, both lateral and axial resolution are improved when using camera-based detection over point-based detection. In terms of axial resolution, we show this in Fig. R9 by comparing axial xz-cross-sections in *Pinus radiata* tissue. Fig. R9a shows the xz-cross-section when using PMT-2PM, while xz-cross-sections acquired with WiL-2PM and LiL-2PM are depicted in Fig. R9b-c, respectively. The corresponding axial resolutions are 648 ± 32 nm for PMT-2PM, 562 ± 27 nm for WiL-2PM and 508 ± 24 nm for LiL-2PM, respectively.

In response to this comment, we added a sentence to the main manuscript stating that we analyzed the axial resolution corresponding to each modality and that this information is shown in section 6 (Fig. S6) in the supplemental information file.

Fig. R9: Comparison of xz-cross sections in *Pinus radiata*, imaged with a) PMT-2PM, b) WiL-2PM and c) LiL-2PM. The axial resolution is increased in the camera-based schemes.

(5) Despite the use of cylindrical lenses reducing the optical complexity compared with adaptive optics, the benefit of low-cost and reduced complexity in LiL-SIM is not justified in the context of improving the scientific community or research. It would still require dedicated two-photon SIM training to implement the system. From a technology dissemination perspective, I suggest the authors to discuss and/or demonstrate how the low-cost and easy implementation features can directly help biological users who are interested in using two-photon SIM.

We thank the reviewer for this comment. As requested, we have discussed these aspects in more detail and would like to refer the reviewer to our answers to point 1 and 3 in our response to reviewer 2.

- [1] Chen B, Chang BJ, Roudot P, Zhou F, Sapoznik E, Marlar-Pavey M, et al. Resolution doubling in light-sheet microscopy via oblique plane structured illumination. *Nat Methods* 2022;19:1419–26. <https://doi.org/10.1038/s41592-022-01635-8>.

- [2] Stuurman N, Amdodaj N, Vale R. μ Manager: Open Source Software for Light Microscope Imaging. *Micros Today* 2007;15:42–3. <https://doi.org/10.1017/S1551929500055541>.
- [3] Ingaramo M, York AG, Wawrzusin P, Milberg O, Hong A, Weigert R, et al. Two-photon excitation improves multifocal structured illumination microscopy in thick scattering tissue. *Proc Natl Acad Sci U S A* 2014;111:5254–9. <https://doi.org/10.1073/pnas.1314447111>.
- [4] Winter PW, York AG, Nogare DD, Ingaramo M, Christensen R, Chitnis A, et al. Two-photon instant structured illumination microscopy improves the depth penetration of super-resolution imaging in thick scattering samples. *Optica* 2014;1:181. <https://doi.org/10.1364/optica.1.000181>.
- [5] Richards B, Wolf E. Electromagnetic diffraction in optical systems, II. Structure of the image field in an aplanatic system. *Proc R Soc Lond A Math Phys Sci* 1959;253:358–79. <https://doi.org/10.1098/rspa.1959.0200>.
- [6] Dorn R, Quabis S, Leuchs G. The focus of light—linear polarization breaks the rotational symmetry of the focal spot. *J Mod Opt* 2003;50:1917–26. <https://doi.org/10.1080/09500340308235246>.
- [7] Diaspro A, Federici F, Robello M. Influence of refractive-index mismatch in high-resolution three-dimensional confocal microscopy. 2002.
- [8] Demmerle J, Innocent C, North AJ, Ball G, Müller M, Miron E, et al. Strategic and practical guidelines for successful structured illumination microscopy. *Nat Protoc* 2017;12:988–1010. <https://doi.org/10.1038/nprot.2017.019>.
- [9] Sahu P, Mazumder N. Improving the Way We See: Adaptive Optics Based Optical Microscopy for Deep-Tissue Imaging. *Front Phys* 2021;9. <https://doi.org/10.3389/fphy.2021.654868>.
- [10] Heintzmann R, Huser T. Super-Resolution Structured Illumination Microscopy. *Chem Rev* 2017;117:13890–908. <https://doi.org/10.1021/acs.chemrev.7b00218>.
- [11] Berning S, Willig KI, Steffens H, Dibaj P, Hell SW. Nanoscopy in a living mouse brain. *Science* (1979) 2012;335:551. <https://doi.org/10.1126/science.1215369>.
- [12] Turcotte R, Liang Y, Tanimoto M, Zhang Q, Li Z, Koyama M, et al. Dynamic super-resolution structured illumination imaging in the living brain. *Proc Natl Acad Sci U S A* 2019;116:9586–91. <https://doi.org/10.1073/pnas.1819965116>.
- [13] Dani A, Huang B, Bergan J, Dulac C, Zhuang X. Superresolution Imaging of Chemical Synapses in the Brain. *Neuron* 2010;68:843–56. <https://doi.org/10.1016/j.neuron.2010.11.021>.
- [14] Herrmannsdörfer F, Flottmann B, Nanguneri S, Venkataramani V, Horstmann H, Kuner T, et al. 3D d STORM Imaging of Fixed Brain Tissue. *Methods in Molecular Biology* 2017;1538:169–84. https://doi.org/10.1007/978-1-4939-6688-2_13.
- [15] Dertinger T, Colyer R, Iyer G, Weiss S, Enderlein J. Fast, background-free, 3D super-resolution optical fluctuation imaging (SOFI). *Proceedings of the National Academy of Sciences* 2009;106:22287–92. <https://doi.org/10.1073/pnas.0907866106>.
- [16] Dertinger T, Xu J, Naini OF, Vogel R, Weiss S. SOFI-based 3D superresolution sectioning with a widefield microscope. *Opt Nanoscopy* 2012;1:1–5. <https://doi.org/10.1186/2192-2853-1-2>.
- [17] Vaziri A, Tang J, Shroff H, Shank C V. Multilayer three-dimensional super resolution imaging of thick biological samples. vol. 23. 2008.

- [18] Schueder F, Lara-Gutiérrez J, Beliveau BJ, Saka SK, Sasaki HM, Woehrstein JB, et al. Multiplexed 3D super-resolution imaging of whole cells using spinning disk confocal microscopy and DNA-PAINT. *Nat Commun* 2017;8. <https://doi.org/10.1038/s41467-017-02028-8>.
- [19] Velasco MGM, Zhang M, Antonello J, Yuan P, Allgeyer ES, May D, et al. 3D super-resolution deep-tissue imaging in living mice. *Optica* 2021;8:442. <https://doi.org/10.1364/optica.416841>.
- [20] Bancelin S, Mercier L, Murana E, Nägerl UV. Aberration correction in stimulated emission depletion microscopy to increase imaging depth in living brain tissue. *Neurophotonics* 2021;8. <https://doi.org/10.1117/1.nph.8.3.035001>.
- [21] Winter PW, Shroff H. Faster fluorescence microscopy: Advances in high speed biological imaging. *Curr Opin Chem Biol* 2014;20:46–53. <https://doi.org/10.1016/j.cbpa.2014.04.008>.
- [22] Schermelleh L, Ferrand A, Huser T, Eggeling C, Sauer M, Biehlmaier O, et al. Super-resolution microscopy demystified. *Nat Cell Biol* 2019;21:72–84. <https://doi.org/10.1038/s41556-018-0251-8>.
- [23] Godin AG, Lounis B, Cognet L. Super-resolution microscopy approaches for live cell imaging. *Biophys J* 2014;107:1777–84. <https://doi.org/10.1016/j.bpj.2014.08.028>.

We want to thank the reviewers again for their positive feedback and their valuable suggestions. Please find our answers to the suggested revisions below.

Reviewer 2:

The description ‘entire frame’ is misleading because 5 ms should refer to the exposure time per line. The total exposure time of the entire frame is closer to $5\text{ ms} + 1024 \times 0.714286\text{ ms} = 736.43\text{ ms}$. Please modify the description accordingly if our description is more accurate.

We thank the reviewers for pointing this out. The first line starts at time zero, so the frame acquisition time should be equal to $5\text{ ms} + 1023 \times 0.714286\text{ ms}$. We modified the sentence to: By setting the exposure time per line to 5~ms, the “light sheet mode line time” is automatically set to $714.286\text{ }\mu\text{s}$ (5~ms~/~7) by Micro-Manager, which amounts to a total exposure time of $5\text{ ms} + 1023 \times 0.714286\text{ ms} = 735.71\text{ ms}$. We clarified this when mentioning line times and frame acquisition times throughout the manuscript, as well as in the supplemental information file.

The hardware timing diagram in SI is good addition (Fig S1), but we do not see the associated caption in the SI file

In our version, the caption is included in the SI file.

In response to our second comment about the choice of fluorescent markers, the authors claim in their rebuttal that they discuss these limitations in the Materials and Methods section of their manuscript. We did not see any discussion here, perhaps we missed it?

Added (lines 419-421): “This is particularly important when using fluorescent markers with low two-photon absorption cross-sections, which somewhat limits the choice of fluorophores.”

It appears that the caption is missing ‘a)’ before ‘WiL-2PM’ in the first sentence of the caption. Also, it would be helpful to indicate by how much the FOV is increased, as mentioned in the caption, relative to the original 100x lens.

Added a) to the figure description. Added the corresponding FOVs in Figure S5. Added (Line 170: see Fig. S4 and S5). Added (Suppl. Caption Figure S5: This corresponds to a maximum FOV of $111 \times 111\text{ }\mu\text{m}$ for the 60x objective lens (compared to a maximum FOV of $67 \times 67\text{ }\mu\text{m}$ for the 100x.)

We appreciate the additional clarity on the mechanism of stepwise line modulation. However, the Galvo voltage ramp shown in Figure S1 b) still may mislead the readers into thinking that the scanner is operating continuously rather than in discrete steps. Thus, we suggest adding a zoomed in figure/inset to illustrate that the input voltage to the scanner is actually operated stepwise.

Modified Fig. S1 and included discrete voltage ramps (shown in the Figure below). Added (Suppl. Lines 25-26): Voltage ramps are generated with discrete increments, which leads to stepwise scanning via the non-resonant galvo-scanner.

Figure S1 a) Hardware diagram and b) timing diagram of the LiL-SIM setup.

We recommend additionally including the point mentioned in the authors' rebuttal, that modulating the intensity of the line during the scan would improve modulation contrast- perhaps a phrase to this effect could be included at the end of section 4.1, where the authors point out that modulation of the laser would complicate synchronization, but don't point out the improvement in contrast that is likely to be achieved by modulating the intensity of the laser.

Modified (Lines 457-460): Since the line exposure times is rather long (2-5 ms) compared to the step time (approx. 300 μ s), we assume that on/off modulation of the laser during the scanning step could likely lead to an improvement in modulation contrast but would only complicate the synchronization.

In response to our point 11, about reducing laser power, the comments in the rebuttal letter make sense. However, in the corresponding text in the manuscript, they appear to compare to 'widefield illumination required for 1-photon excited SIM...' We suspect that widefield 1-photon SIM requires far less power than applied here. Perhaps the authors should rephrase to something like, '..in comparison to the power required for full FOV 2p illumination, this also reduces the required laser power...'

Modified (Lines 166-168): Furthermore, in comparison to the laser power required for full FOV 2P-illumination, this also reduces the required power by a factor given by the number of lines that make up the final pattern (up to 200).

-Fig2, we appreciate the changes and associated careful SNR/SBR analysis. Please specify in the caption/figure what the sample is in Fig. 2a-c, and indicate in the caption that the sample in Fig. 2a is distinct from that shown in Fig. 2b, c.

Modified (Figure 2 Caption) Comparison of signal-to-background and signal-to-noise ratio in two-photon fluorescence microscopy images acquired in *Pinus radiata* tissue with a) photomultiplier tube (PMT), b) camera with rolling shutter (RS) mode, and c) camera with lightsheet shutter (LSS) mode. The images presented in a) were acquired with a commercial PMT-based microscope system at the same imaging depth ($z = 10 \mu\text{m}$) but at distinct lateral positions of the specimen.

Reviewer 3:

The authors have partially addressed my concerns. However, since the content presented in the manuscript largely focuses on the accessibility and cost-efficiency other than pushing the technological limit. I still felt the manuscript does not meet the scope of Nature Communication considering the comparable imaging penetration, speed and resolution metrics achieved by similar methods. Also, considering the cost reduction, the outlined number of optical parts are around ~10.000-20.000 Euros, and I felt this level of reduction from ~30.000-40.000 Euros using EMCCD is relatively marginal. Therefore, I recommend considering publishing it in a more specialized journal such as Optica or BOE.

We appreciate the reviewer's comment on pricing, which indicates that our statement was not entirely clear. As mentioned in our manuscript, other 2P-SIM setups require not only an EMCCD camera, but also major modifications to the setup at additional cost. In contrast, our complete setup costs ~20,000 euros - almost half the price of the EMCCD camera alone. This significant price reduction, combined with off-the-shelf hardware and simplified implementation, suggests that our technique could be of great benefit to the microscopy community and facilitate access to super-resolution imaging in smaller laboratories.